# A *Ctnnb1* enhancer transcriptionally regulates Wnt signaling dosage to balance homeostasis and tumorigenesis of intestinal epithelia

Xiaojiao Hua[1,2], Chen Zhao[1,2], Jianbo Tian[3,4], Junbao Wang[1,2], Xiaoping Miao[3,4], Gen Zheng[5], Min Wu[2,4,6], Mei Ye[7], Ying Liu[1,2]*, Yan Zhou[1,2]*

[1]Department of Neurosurgery, Medical Research Institute, Zhongnan Hospital of Wuhan University, Wuhan University, Wuhan, China; [2]Frontier Science Center of Immunology and Metabolism, Wuhan University, Wuhan, China; [3]Department of Epidemiology and Biostatistics, School of Public Health, Wuhan University, Wuhan, China; [4]TaiKang Center for Life and Medical Sciences, Wuhan University, Wuhan, China; [5]Department of Gastroenterology, Union Hospital, Tongji Medical College, Huazhong University of Science and Technology, Wuhan, China; [6]College of Life Sciences, Wuhan University, Wuhan, China; [7]Department of Gastroenterology, Zhongnan Hospital of Wuhan University, Wuhan, China

*For correspondence:
y.liu@whu.edu.cn (YL);
yan.zhou@whu.edu.cn (YZ)

**Competing interest:** The authors declare that no competing interests exist.

**Abstract** The β-catenin-dependent canonical Wnt signaling is pivotal in organ development, tissue homeostasis, and cancer. Here, we identified an upstream enhancer of *Ctnnb1* – the coding gene for β-catenin, named ieCtnnb1 (*i*ntestinal *e*nhancer of *Ctnnb1*), which is crucial for intestinal homeostasis. ieCtnnb1 is predominantly active in the base of small intestinal crypts and throughout the epithelia of large intestine. Knockout of ieCtnnb1 led to a reduction in *Ctnnb1* transcription, compromising the canonical Wnt signaling in intestinal crypts. Single-cell sequencing revealed that ieCtnnb1 knockout altered epithelial compositions and potentially compromised functions of small intestinal crypts. While deletion of ieCtnnb1 hampered epithelial turnovers in physiologic conditions, it prevented occurrence and progression of Wnt/β-catenin-driven colorectal cancers. Human ieCTNNB1 drove reporter gene expression in a pattern highly similar to mouse ieCtnnb1. ieCTNNB1 contains a single-nucleotide polymorphism associated with *CTNNB1* expression levels in human gastrointestinal epithelia. The enhancer activity of ieCTNNB1 in colorectal cancer tissues was stronger than that in adjacent normal tissues. HNF4α and phosphorylated CREB1 were identified as key trans-factors binding to ieCTNNB1 and regulating *CTNNB1* transcription. Together, these findings unveil an enhancer-dependent mechanism controlling the dosage of Wnt signaling and homeostasis in intestinal epithelia.

## eLife assessment

Ctnnb1 encodes β-catenin, an essential component of the canonical Wnt signaling pathway. In this **important** study, the authors identify an upstream enhancer of Ctnnb1 responsible for the specific expression level of β-catenin in the gastrointestinal track. Deletion of this enhancer in mice and analyses of its association with human colorectal tumors provide **compelling** support that it controls the dosage of Wnt signaling critical to the homeostasis in intestinal epithelia and colorectal cancers.

## Introduction

The gastrointestinal (GI) epithelium, a high turnover organ, contains multiple cell types to fulfill complex functions including food digestion, nutrient absorption, pathogen insulation, and clearance, as well as endocrine roles (*Noah et al., 2011*; *Chin et al., 2017*; *Kurokawa et al., 2020*). Enterocytes (ECs) constitute ~90% of all differentiated cells and function as highly efficient cells for absorbing nutrients and water (*Snoeck et al., 2005*). Goblet cells (GCs) produce mucus to lubricate the mucosal surface (*Gustafsson and Johansson, 2022*). Enteroendocrine cells (EECs) sense luminal nutrients and secrete peptide hormones to regulate appetite, intestinal motility, and insulin release (*Gribble and Reimann, 2019*). Tuft cells (TCs) are chemosensory cells that respond to various stimuli including infections and hypoxia (*Schneider et al., 2019*). The crypts of GI epithelium harbor various types of stem cells that continually replenish lost epithelial cells (*van der Flier and Clevers, 2009*; *Ramadan et al., 2022*). While the prevailing belief is that *Lgr5*-expressing crypt base columnar (CBC) cells are actively dividing stem cells and *Bmi*-expressing '+4' cells (positioned four cells above the base of the crypt) are slowly cycling stem cells, it's important to note that crypt cells have the capacity for fate reprogramming and de-differentiation to sustain regenerative capability (*Barker et al., 2007*; *Tian et al., 2011*; *Yan et al., 2012*; *Metcalfe et al., 2014*; *Beumer and Clevers, 2016*). Intestinal stem cells generate transit-amplifying cells (TACs), which undergo multiple cell divisions and differentiate into all intestinal lineages while migrating toward the crypt orifice (*Krausova and Korinek, 2014*). In addition to stem and progenitor cells, the base of small intestinal crypts contains Paneth cells (PCs) that secret antimicrobial peptides, called defensins, thereby contributing to host defense against microbes (*Lueschow and McElroy, 2020*; *Barreto et al., 2022*). PCs also contribute to the stem cell niche function, e.g., secreting Wnt ligands, to maintain epithelial homeostasis of the small intestine (*Sato et al., 2011*).

The behavior of crypt cells is precisely regulated by multiple signals, which can act sequentially, additively, synergistically, or antagonistically (*Nakamura et al., 2007*). Among them, the Wnt/β-catenin signal pathway represents the principal force governing intestinal epithelium homeostasis, particularly in the preservation of stem cell proliferation and multipotency (*Pinto et al., 2003*; *Clevers et al., 2014*; *Mah et al., 2016*). Upon Wnt ligand binding, the β-catenin degradation complex is inactivated, leading to accumulation of β-catenin and its translocation into the nucleus, where it works with the co-activator TCF7L2 to turn on the transcription of a plethora of target genes, including *MYC*, *CCND1*, *AXIN2*, and *CD44* (*Clevers, 2006*; *Clevers and Nusse, 2012*; *Nusse and Clevers, 2017*). Inhibition of Wnt ligand secretion reduced the number of functional stem cells and led to a faster crypt fixation rate (*Huels et al., 2018*). In the process of aging, reduced Wnt signaling in stem cells and the niche of intestinal crypts causes inhibition of proliferation and reduced numbers of intestinal stem cells (*Nalapareddy et al., 2017*; *Pentinmikko et al., 2019*). The Wnt signal is also essential for commitment, maturation, and location of PCs and other secretory cells (*van Es et al., 2005*; *Andreu et al., 2008*). On the other hand, Wnt, EGF, and Delta-like 1/4 ligands presented by PCs sustained CBCs' proliferative states and the production (*Sato et al., 2011*). Although the Wnt signal dosage, largely via post-translational regulations, controls early lineage specification, hematopoiesis, and tumorigenesis (*Luis et al., 2011*; *Bakker et al., 2013*), it remains elusive how Wnt signaling dosage, particularly through the transcriptional control of *Ctnnb1* – the coding gene for β-catenin, regulates composition and function of intestinal epithelia.

Aberrant activation of canonical Wnt signaling is associated with GI cancers, most notably colorectal cancer (CRC) (*Zhao et al., 2022*; *Zhu and Li, 2023*). Germinal mutations of the *APC* gene, which encodes the negative regulator of β-catenin, underlie the familial adenomatous polyposis, a hereditary neoplastic syndrome and the pre-cancerous condition for CRC (*Cheng et al., 2019*; *Li et al., 2021a*). Moreover, *APC* is mutated in up to 80% of human sporadic CRCs (*Li et al., 2021b*). Inactivation of APC causes enhanced accumulation of β-catenin in nuclei and subsequent activation of proliferative and pro-survival signals in CRC cells (*Cheng et al., 2019*). However, largely due to lack of specificity, no molecular therapeutic strategy targeting the pathway has been incorporated into CRC treatment. Interestingly, the transcriptional activity of *CTNNB1*, the coding gene for β-catenin, is also enhanced in CRCs (*Terrin et al., 2017*); however, the underlying mechanism is largely unknown. Dissecting it would facilitate the development of new Wnt/β-catenin targeting means to treat CRC.

Enhancers are essential *cis*-regulatory DNA elements that control spatiotemporal gene expressions during cell fate specification and maintenance (*Pachano et al., 2022*). Active enhancers are often associated with cell fate specifiers, mostly transcription factors (TFs), and are enriched with specific

histone modification such as H3K27ac (acetylated lysine 27 of histone H3) and H3K4me1 (mono-methylated lysine 4) of histone H3 (*Jindal and Farley, 2021*). Importantly, variations of enhancer sequences often lead to aberrant gene silence and activation, thereby causing developmental anomalies and cancers (*Zabidi and Stark, 2016*; *Sengupta and George, 2017*; *Kvon et al., 2021*). Although numerous putative enhancers have been identified by high-throughput studies, only a fraction of them were functionally annotated (*Arnold et al., 2013*; *Li et al., 2021c*; *Yao et al., 2022*).

The proper expression of essential fate genes in different cell types is often controlled by specific enhancers (*Kvon et al., 2021*; *Pachano et al., 2022*). Here, we have identified an intestinal enhancer for *Ctnnb1* and name it as ieCtnnb1 (intestinal enhancer of *Ctnnb1*). Reporter mice showed that ieCtnnb1 is predominantly active in the base of small intestinal crypts and across the epithelium of large intestine. Knockout of ieCtnnb1 slows down turnovers of intestinal epithelia at physiology and cancer conditions in a Wnt/β-catenin-dependent manner. The human ieCTNNB1 contains a single-nucleotide polymorphism (SNP) that is associated with *CTNNB1* expression levels in human GI epithelia. The enhancer activity of ieCTNNB1 in CRC tissues is higher than that in adjacent normal tissues and positively correlates with *CTNNB1* expression levels. HNF4α and phosphorylated CREB1 at serine 133 (p-S133-CREB1) were found to associate with ieCtnnb1 and regulate *Ctnnb1* transcription.

## Results

### ieCtnnb1 is a GI enhancer of *Ctnnb1*

We previously identified an evolutionarily conserved upstream enhancer for *Ctnnb1* – neCtnnb1, which regulates neocortical neurogenesis (*Figure 1—figure supplement 1A*). We reasoned that there might be other potential enhancer regions that control *Ctnnb1* expression in a cell-specific manner. Therefore, we scanned upstream regions of *Ctnnb1* for active enhancer properties – open chromatin with enriched H3K27ac and H3K4me1 – in different tissues using data retrieved from the ENCODE database. Interestingly, we noticed that a region that is 29,068 base pair (bp) upstream of the transcription starting site for *Ctnnb1* displayed typical enhancer feature in intestinal tissues and resides in the same topologically associating domain (TAD) with *Ctnnb1* (*Figure 1A*, *Figure 1—figure supplement 2A*). The region, named as ieCtnnb1 (intestinal enhancer of *Ctnnb1*), is at open chromatin status (DNase I hypersensitive site) and enriched with H3K27ac and H3K4me1, two histone marks for active enhancers, in 8-week-old mouse small intestine tissue (*Figure 1A*). Importantly, the enrichment of H3K27ac at ieCtnnb1 is also prominent in embryonic intestine and stomach tissue, as well as in adult kidney and liver (*Figure 1—figure supplement 1B and D*). In contrast, the enrichment was not detected in non-GI tissues such as embryonic and adult brain (*Figure 1—figure supplement 1D*). In line with signatures of active enhancers, the enrichment of H3K4me3 and H3K36me3 at ieCtnnb1 is negligible in all examined tissues (*Figure 1A*, *Figure 1—figure supplement 1B and C*). Notably, unlike neCtnnb1, the primary sequence of ieCtnnb1 is not conserved among vertebrates (*Figure 1A*, bottom).

We next performed in vitro and in vivo analyses to validate ieCtnnb1's enhancer activity. First, the 2,153 bp core region (CR), representing the H3K27ac peak region of ieCtnnb1, could drive the expression of luciferase reporter in multiple cell types (*Figure 1—figure supplement 2B*). Second, the ieCtnnb1 reporter mice ($H11^{i.enh}$) were constructed by inserting the reporter cassette – 5'-*Shh*'s promoter-ieCtnnb1-CR-*LacZ*-iCre-3' – into the *H11* locus (*Figure 1B*). Of note, the *Shh* promoter alone at the *H11* locus did not drive the expression of reporter gene in all examined tissue (*Kvon et al., 2020*). Whole-mount β-galactosidase (β-gal) staining of 8-week-old tissues revealed prominent ieCtnnb1-*LacZ* signals along the GI tract. The signal exhibited a proximal-low to distal-high gradient, with the ileum, cecum, and colon showing the strongest signals (*Figure 1B*, *Figure 1—figure supplement 2C*). The stomach and proximal duodenum also displayed reporter activity driven by ieCtnnb1 (*Figure 1B*, *Figure 1—figure supplement 2C*). β-Gal staining of intestinal sections revealed that the ieCtnnb1-*LacZ* signal could be visualized at the bottom of small intestinal crypts and in the entire large intestinal epithelia, but is absent in sections of liver, kidney, and spleen (*Figure 1—figure supplement 2D* and data not shown). Interestingly, the proximal-low to distal-high gradient of ieCtnnb1-*LacZ* signal along the small intestine corresponds to the expression pattern of β-catenin (*Figure 1—figure supplement 2E*).

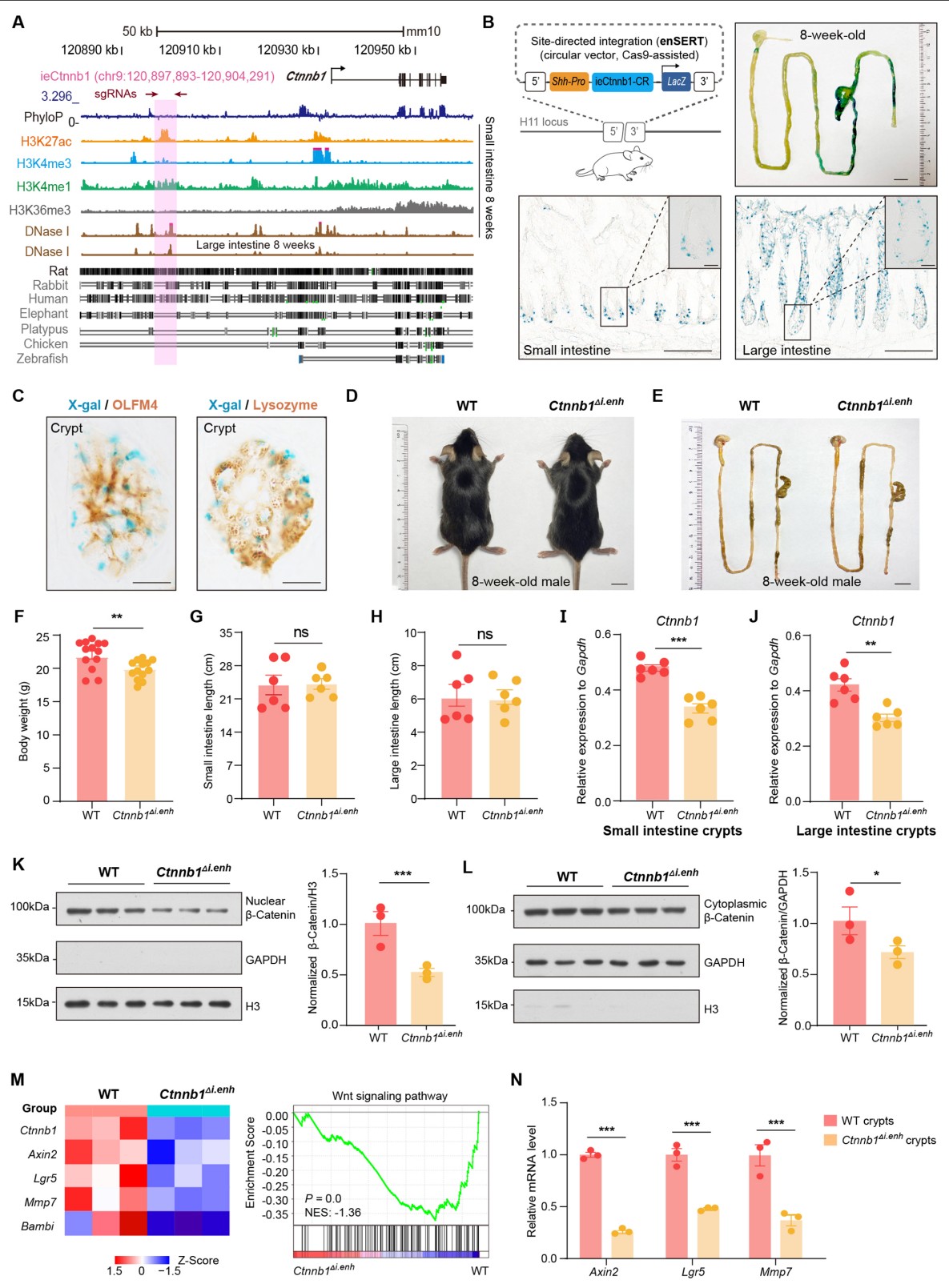

**Figure 1.** ieCtnnb1 is an intestinal enhancer of *Ctnnb1*. (**A**) Schematic representation of the upstream region of mouse *Ctnnb1* gene and the location of ieCtnnb1 (6,399 bp, pink shading), which is marked by H3K27ac and H3K4me1 peaks, and DNase I hypersensitivity in small intestine and large intestine of 8-week-old mice. The sequence conservation of the indicated species is shown at the bottom as vertical lines. Data were obtained from ENCODE. Locations of single-guide RNAs (sgRNAs) for generating ieCtnnb1 knockout mice were marked. (**B**) Top left: a schematic illustration showing that the

*Figure 1 continued on next page*

*Figure 1 continued*

knock-in reporter construct carries the *Shh* promoter, ieCtnnb1 core region sequences (2,153 bp), and the *LacZ* reporter gene. Top right: X-Gal staining (blue) of the gastrointestinal (GI) tract of an 8-week-old *H11$^{i.enh}$* mouse. Bottom: X-Gal staining (blue) of the small intestine (left) and colon (right) of an 8-week-old *H11$^{i.enh}$* mouse. Boxed areas were enlarged at top-right corners. (**C**) Representative images of small intestinal crypts co-labeled by X-Gal with OLFM4 (left), and X-Gal with Lysozyme (right). (**D–E**) Representative images of whole body (**d**) and GIs (**e**) of 8-week-old male wildtype (WT) and *Ctnnb1$^{Δi.enh}$* mice. (**F**) Comparison of the body weight of 8-week-old male WT (n=13) and *Ctnnb1$^{Δi.enh}$* (n=13) mice. (**G–H**) Measurements of small (**G**) and large (**H**) intestine length of 8-week-old male WT (n=6) and *Ctnnb1$^{Δi.enh}$* (n=6) mice. (**I–J**) Relative mRNA levels of *Ctnnb1* in small (**I**) and large (**J**) intestinal crypts of WT (n=6) and *Ctnnb1$^{Δi.enh}$* (n=6) mice. (**K–L**) Left: immunoblotting of nuclear (**K**) and cytoplasmic (**L**) β-catenin, GAPDH, and H3 of small intestinal crypts of WT (n=3) and *Ctnnb1$^{Δi.enh}$* (n=3) mice. Right: histograms showing protein levels of β-catenin normalized to H3 (**K**) or GAPDH (**L**) levels. Values of WT were set as '1'. (**M**) Heatmap of indicated Wnt target genes and gene set enrichment analysis (GSEA) of Wnt signaling pathway according to transcriptome profiles of small intestinal crypts of WT (n=3) and *Ctnnb1$^{Δi.enh}$* (n=3) mice. (**N**) Quantitative reverse transcription PCR (RT-qPCR) showing relative mRNA levels of indicated Wnt target genes (*Axin2, Lgr5,* and *Mmp7*) in small intestinal crypts of WT (n=3) and *Ctnnb1$^{Δi.enh}$* (n=3) mice. Scale bars, 1 cm (B, top; **D and E**), 100 μm (B, bottom), 10 μm (B, magnified views; **C**). Quantification data are shown as means ± SEM, statistical significance was determined using an unpaired two-tailed Student's *t*-test (**F–L**). Quantification data are shown as means ± SD, statistical significance was determined using Multiple *t*-tests – one per row (**N**). *p<0.05, **p<0.01, ***p<0.001, and ****p<0.0001. ns, not significant. NES: normalized enrichment score.

The online version of this article includes the following source data and figure supplement(s) for figure 1:

**Source data 1.** Numerical data for *Figure 1F*.

**Source data 2.** Numerical data for *Figure 1G*.

**Source data 3.** Numerical data for *Figure 1H*.

**Source data 4.** Numerical data for *Figure 1I*.

**Source data 5.** Numerical data for *Figure 1J*.

**Source data 6.** Numerical data for *Figure 1K*.

**Source data 7.** Numerical data for *Figure 1L*.

**Source data 8.** Numerical data for *Figure 1N*.

**Source data 9.** Uncropped and labeled gels for *Figure 1*.

**Source data 10.** Raw unedited gels for *Figure 1*.

**Figure supplement 1.** ieCtnnb1 is a putative intestinal enhancer upstream of *Ctnnb1*.

**Figure supplement 2.** ieCtnnb1 is predominantly active in developing intestine.

**Figure supplement 2—source data 1.** Numerical data for *Figure 1—figure supplement 2B*.

**Figure supplement 2—source data 2.** Numerical data for *Figure 1—figure supplement 2F*.

**Figure supplement 2—source data 3.** Numerical data for *Figure 1—figure supplement 2J*.

**Figure supplement 2—source data 4.** Numerical data for *Figure 1—figure supplement 2K*.

**Figure supplement 2—source data 5.** Uncropped and labeled gels for *Figure 1—figure supplement 2*.

**Figure supplement 2—source data 6.** Raw unedited gels for *Figure 1—figure supplement 2*.

**Figure supplement 3.** The list of Wnt signaling pathway components downregulated in *Ctnnb1$^{Δi.enh}$* crypts.

The transcription activity of ieCtnnb1 is strongest at the bottom of crypts, where *Lgr5*-expressing CBC cells and PCs reside. To more precisely elucidate cell types in which ieCtnnb1 is active, we crossed *H11$^{i.enh}$* mice with *Lgr5*-EGFP-IRES-CreERT2 (*Lgr5*-EGFP) mice to obtain 8-week-old *H11$^{i.enh}$;Lgr5-EGFP* mice. Small intestinal crypts were respectively harvested from *Lgr5-EGFP* and *H11$^{i.enh}$;Lgr5-EGFP* mice followed by enrichment of both eGFP+ and eGFP- cells (*Figure 1—figure supplement 2F*). Quantitative reverse transcription PCR (RT-qPCR) analyses indicated that *LacZ* transcripts could be detected in both eGFP+ and eGFP- crypt cells in *H11$^{i.enh}$;Lgr5-EGFP* mice (*Figure 1—figure supplement 2F*). Next, sections of *H11$^{i.enh}$* crypts were subjected with β-gal staining followed by immunohistochemistry of OLFM4 (a CBC marker) or lysozyme (a PC marker). Data showed X-gal signals are in close proximity with both OLFM4 and lysozyme (*Figure 1C*), suggesting that the activity of ieCtnnb1 is present in both CBCs and PCs.

We further examined the ieCtnnb1 activity in GI tracts of postnatal day 7 (P7) pups, when GI epithelia are rapidly expanding. The ieCtnnb1-*LacZ* signals could be visualized in epithelia of P7 small and large intestines, aligning well with the distribution of canonical Wnt signaling illustrated by the BAT-Gal mice (*Figure 1—figure supplement 2G and H*). Together, ieCtnnb1 drives transcription in embryonic intestinal epithelia, the base of small intestinal crypts, and large intestinal epithelia of adult mice.

## ieCtnnb1 knockout decreased the expression of *Ctnnb1* in intestinal epithelia

We next examined whether ieCtnnb1 controls the transcription of *Ctnnb1* in intestinal epithelia. For this purpose, we employed CRISPR/Cas9-mediated gene editing to delete the genomic region containing ieCtnnb1 (*Figure 1—figure supplement 2I*). Homozygous ieCtnnb1 knockout (*Ctnnb1$^{\Delta i.enh}$*) mice were born in Mendelian ratios and thrived through adulthood. Interestingly, the body weight of 8-week-old male *Ctnnb1$^{\Delta i.enh}$* mice was slightly but significantly lighter than that of wild-type controls (*Figure 1D and F*), but the length of small and large intestines of *Ctnnb1$^{\Delta i.enh}$* mice was comparable to that of control mice (*Figure 1E, G, and H*). We then collected crypts from small and large intestine, finding that the expression levels of *Ctnnb1* were decreased by 26.5% and 22.3% respectively in *Ctnnb1$^{\Delta i.enh}$* crypts (*Figure 1I and J*). In contrast, the expression level of *Ctnnb1* in the *Ctnnb1$^{\Delta i.enh}$* liver remained unaltered (*Figure 1—figure supplement 2J*). In addition, the expression of genes located in the same TAD as *Ctnnb1* and ieCtnnb1, including *Ulk4*, *Rpl14*, and *Entpd3*, was not changed in *Ctnnb1$^{\Delta i.enh}$* small intestinal crypts (*Figure 1—figure supplement 2K*). These data suggest that ieCtnnb1 plays a specific role in regulating the transcription of *Ctnnb1* in intestinal epithelia.

In epithelial tissues, most β-catenin is localized at cell adhesion sites to maintain barrier integrity, while a smaller fraction translocates into the nucleus to activate transcription upon Wnt ligand binding. We therefore carried out nuclear/cytosol fractionation assays, which revealed a substantial decrease in the levels of both nuclear (49.5%) and cytosolic (29.8%) β-catenin in small intestinal crypts of *Ctnnb1$^{\Delta i.enh}$* mice (*Figure 1K and L*). RNA-seq transcriptome and RT-qPCR analyses revealed a significant reduction in the expressions of Wnt target genes in *Ctnnb1$^{\Delta i.enh}$* crypts. Additionally, the gene set enrichment analysis (GSEA) showed that the expression levels of many components of the Wnt signaling pathway were also downregulated (*Figure 1M and N*, *Figure 1—figure supplement 3*). Together, the loss of ieCtnnb1 compromised the Wnt signaling in intestinal epithelia by reducing *Ctnnb1* transcription.

## Loss of ieCtnnb1 altered cellular composition and expression profiles of small intestinal crypts

The canonical Wnt/β-catenin signaling is essential for the construction and homeostasis of intestinal epithelia (*Mah et al., 2016*). Because ieCtnnb1 maintains the expression level of *Ctnnb1* of intestinal crypts, we investigated how the deletion of ieCtnnb1 affects cellular components and functions of intestinal crypts. We performed single-cell sequencing on EpCAM+ epithelial cells from small intestinal crypts of 10-week-old *Ctnnb1$^{\Delta i.enh}$* and control (*H11$^{i.enh}$*) mice (n=2 female mice for each genotype) (*Figure 2A*, *Figure 2—figure supplement 1A*; *Ayyaz et al., 2019*; *Huang et al., 2022*). Uniform Manifold Approximation and Projection (UMAP) was employed for dimensionality reduction to cluster cells, then well-established marker genes for intestinal epithelial cells were mapped onto the UMAP representation to designate cell types (*Haber et al., 2017*; *Gu et al., 2022*; *Figure 2B*, *Figure 2—figure supplement 1B–D*). Of note, a small population of crypt cells remains undefined, as we did not exclude non-epithelial cells such as immune cells, erythrocytes, or endothelial cells during the enrichment of EpCAM+ epithelial cells. Additionally, our focus was exclusively on epithelial cell types, and we did not consider other cell types when defining cell populations (*Figure 2—figure supplement 1D*). First, in accordance with bulk RNA-seq and RT-qPCR results, the expression levels of *Ctnnb1* were downregulated in almost all *Ctnnb1$^{\Delta i.enh}$* crypt cells, and the expression of *Axin2* was decreased in CBCs and TACs (*Figure 2C and D*). Second, the proportion of PCs was decreased by 49.2% in *Ctnnb1$^{\Delta i.enh}$* crypts, which was confirmed by the immunohistochemical (IHC) staining of *Defensin 5α*, a PC marker (*Figure 2E and F*). The ieCtnnb1-driving *LacZ* could be detected in the PC cluster (n=four 10-week-old mice) (*Figure 2—figure supplement 1E*). Accordingly, the expression levels of examined PC markers – *Lyz1*, *Pla2g2a*, *Wnt3*, and *Math1* – were all significantly compromised in *Ctnnb1$^{\Delta i.enh}$* crypts (*Figure 2G*). The number of GCs was slightly (not statistically significant) reduced upon loss of ieCtnnb1 (*Figure 2—figure supplement 1F*). Third, the proliferative capability of CSCs and TACs were mildly reduced in *Ctnnb1$^{\Delta i.enh}$* crypts, as the proportion of G2/M cells and the expression of *Mki67* were decreased in the two cell types (*Figure 2H–J*). Consistently, the numbers of Ki67-positive cells and EdU-labeled cells (4 hr pulse) were decreased by 16.1% and 21.9%, respectively, in *Ctnnb1$^{\Delta i.enh}$* small intestinal epithelia (*Figure 2K and L*). Fourth, in secretory cells (PCs, GCs, and EECs), as well as enterocytes/absorptive cells (mature enterocytes [mECs] and immature enterocytes [imECs], and

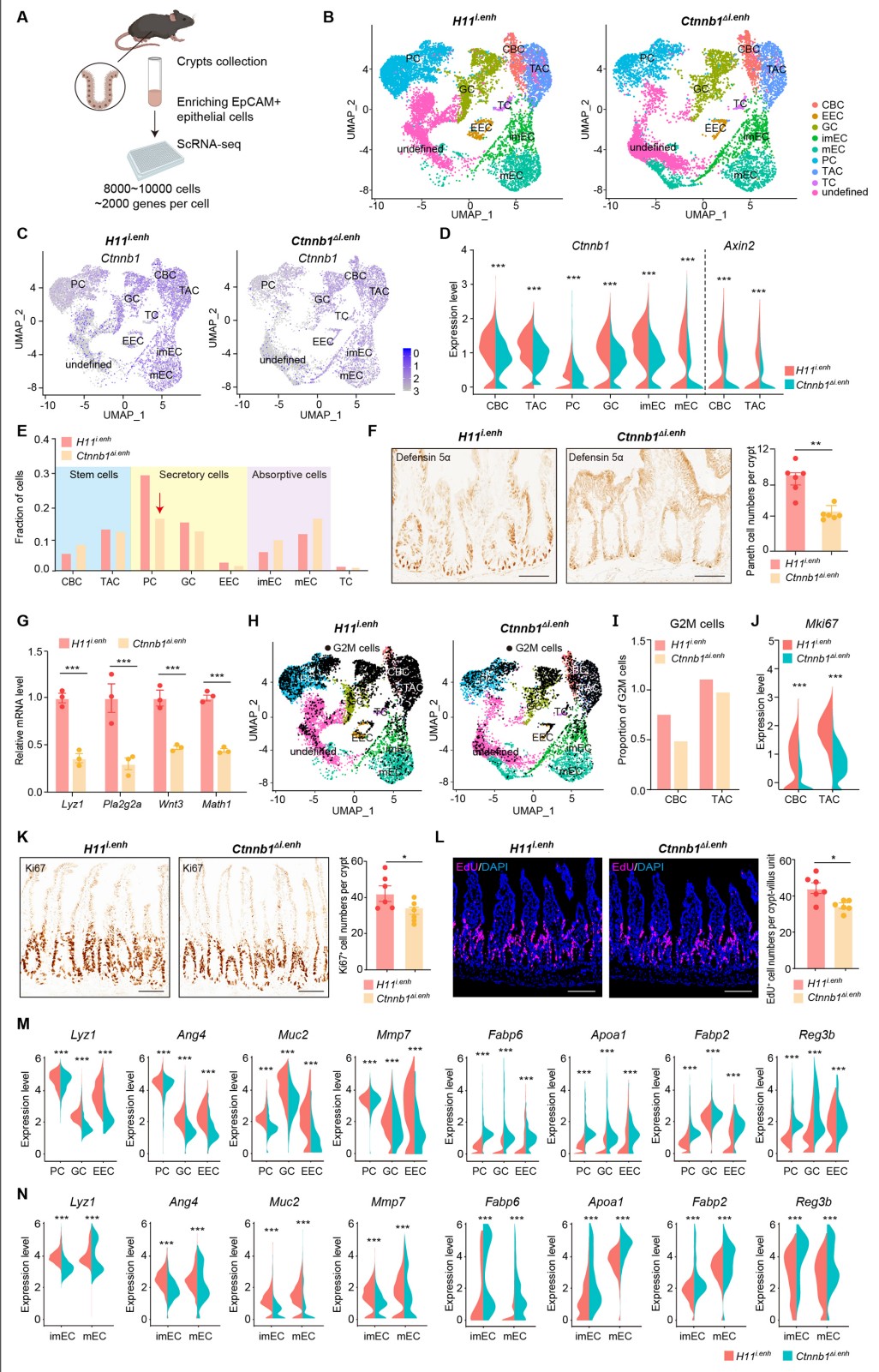

**Figure 2.** ieCtnnb1 knockout altered cellular composition and expression profiles of small intestinal crypts. (**A**) Schematic illustration of single-cell sequencing. Crypts were extracted from small intestines followed by fluorescence activated cell sorting (FACS) to enrich EpCAM+ DAPI- epithelial cells. Cells of two 10-week-old female mice for each genotype were pooled together to perform single-cell transcriptome analyses. (**B**) Uniform

*Figure 2 continued on next page*

*Figure 2 continued*

Manifold Approximation and Projection (UMAP) were used to visualize the clustering of 11,824 single cells from *H11^{i.enh}* mice and 8,094 single cells from *Ctnnb1^{Δi.enh}* mice. Cell types were assigned according to expressions of marker genes. CBC, crypt base columnar cell; TAC, transit-amplifying cell; EEC, enteroendocrine cell; imEC, immature enterocytes; mEC, mature enterocytes; GC, goblet cell; PC, Paneth cell; TC, tuft cell. (**C**) Expression and distribution of *Ctnnb1* in small intestinal crypt cells of *H11^{i.enh}* and *Ctnnb1^{Δi.enh}* mice. (**D**) Violin plots showing the expression of *Ctnnb1* in CBC, TAC, PC, GC, imEC, mEC; and the expression of *Axin2* in CBC and TAC, of *H11^{i.enh}* and *Ctnnb1^{Δi.enh}* mice. (**E**) Comparison of the proportion of indicated small intestinal crypt cell types in *H11^{i.enh}* and *Ctnnb1^{Δi.enh}* mice. (**F**) Immunohistochemistry (left and middle) and quantification (right) of PCs in small intestines of *H11^{i.enh}* (n=6) and *Ctnnb1^{Δi.enh}* (n=6) mice. (**G**) Quantitative reverse transcription PCR (RT-qPCR) showing relative mRNA levels of PC marker genes (*Lyz1, Pla2g2a, Wnt3, Math1*) in small intestinal crypts of *H11^{i.enh}* (n=3) and *Ctnnb1^{Δi.enh}* (n=3) mice. (**H**) Distribution of G2M cells in *H11^{i.enh}* and *Ctnnb1^{Δi.enh}* small intestinal crypts, based on the expression of cell cycle marker gene *Mki67*. (**I**) Comparison of the proportion of G2M cells in CBC and TAC of *H11^{i.enh}* and *Ctnnb1^{Δi.enh}* small intestinal crypts. (**J**) Violin plots showing the expression of *Mki67* in CBC and TAC of *H11^{i.enh}* and *Ctnnb1^{Δi.enh}* small intestinal crypts. (**K**) Immunohistochemistry (left and middle) and quantification (right) of Ki67+ cells in small intestines of *H11^{i.enh}* (n=6) and *Ctnnb1^{Δi.enh}* (n=6) mice. (**L**) Immunofluorescence (left and middle) and quantification (right) of EdU+ cells (red) in small intestines of *H11^{i.enh}* (n=6) and *Ctnnb1^{Δi.enh}* (n=6) mice after 4 hr EdU injection. Nuclei were labeled with DAPI (blue). (**M–N**) Violin plots showing expressions of marker genes for secretory cells (*Lyz1, Ang4, Muc2, Mmp7*) and absorptive cells (*Fabp6, Apoa1, Fabp2, Reg3b*) in secretory(M) and absorptive lineages (N) of *H11^{i.enh}* and *Ctnnb1^{Δi.enh}* small intestinal crypts. Scale bars, 50 μm (F, K, and L). Quantification data are shown as means ± SEM, statistical significance was determined using an unpaired two-tailed Student's *t*-test (**D, F, J, K, L, M, and N**). Quantification data are shown as means ± SD, statistical significance was determined using Multiple *t*-tests – one per row (**G**). *p<0.05, **p<0.01, ***p<0.001, and ****p<0.0001. ns, not significant.

The online version of this article includes the following source data and figure supplement(s) for figure 2:

**Source data 1.** Numerical data for *Figure 2E*.

**Source data 2.** Numerical data for *Figure 2F*.

**Source data 3.** Numerical data for *Figure 2G*.

**Source data 4.** Numerical data for *Figure 2I*.

**Source data 5.** Numerical data for *Figure 2K*.

**Source data 6.** Numerical data for *Figure 2L*.

**Figure supplement 1.** Single-cell survey of small intestinal crypt cells upon ieCtnnb1 knockout.

**Figure supplement 1—source data 1.** Numerical data for *Figure 2—figure supplement 1F*.

stem cells of *Ctnnb1^{Δi.enh}* crypts), the expression of genes related to secretory functions including *Lyz1, Ang4, Muc2*, and *Mmp7* was decreased. In contrast, the expression of markers for absorptive cells such as *Fabp6, Apoa1, Fabp2*, and *Reg3b* was elevated (***Figure 2M and N***, ***Figure 2—figure supplement 1G***). Accordingly, gene ontology (GO) analyses showed in stem cells and secretory cells of *Ctnnb1^{Δi.enh}* crypts, upregulated genes were enriched with terms associated with absorptive functions including ribosome biogenesis and oxidative phosphorylation, whereas downregulated genes in stem cells and secretory cells were respectively enriched with terms related to cell division and secretory functions (***Figure 2—figure supplement 1H and I***). In contrast, in mECs, downregulated genes were enriched with terms associated with absorptive functions (***Figure 2—figure supplement 1J***).

In summary, the loss of ieCtnnb1 results in a reduction in the number of PCs, compromises stem cell proliferation, and disrupts the expression of genes related to secretory and absorptive functions. These effects could be attributed to a combination of reduced Wnt/β-catenin signaling and dysregulated crosstalk among crypt cells.

## ieCtnnb1 deletion inhibits tumorigenesis of intestinal cancers by suppressing Wnt/β-catenin signaling

Tumorigenesis of most CRC is associated with aberrant activation of the Wnt/β-catenin signaling (***Zhu and Li, 2023***). To investigate whether knockout of ieCtnnb1 could suppress the carcinogenesis of Wnt-driven intestinal tumors, we crossbred *Ctnnb1^{Δi.enh}* mice with *Apc^{Min/+}* mice. In *Apc^{Min/+}* mice, a nonsense mutation of the *Apc* gene results in the translation of truncated and dysfunctional APC protein, leading to failed degradation and nuclear translocation of β-catenin. *Apc^{Min/+}* mice

spontaneously develop intestinal tumors due to enhanced canonical Wnt/β-catenin signaling in the intestinal epithelia, particularly in stem cells (*Su et al., 1992*). The survival period of *Ctnnb1^Δi.enh^;Apc^Min/+^* mice (median survival – 255 days) was significantly longer than the *Apc^Min/+^* mice (median survival – 165 days), with the former having a heavier body weight (*Figure 3A and C*). Strikingly, the *Ctnnb1^Δi.enh^;Apc^Min/+^* mice barely grow CRCs and have much fewer intestinal tumors compared to *Apc^Min/+^* mice, with significantly smaller tumor area (*Figure 3B and D–F*, *Figure 3—figure supplement 1A–C*). IHC analyses showed that tumors in *Ctnnb1^Δi.enh^;Apc^Min/+^* large and small intestines had much fewer Ki67+ proliferating cells, expressed significant less β-catenin, but contained more Muc2-expressing GCs, suggesting *Ctnnb1^Δi.enh^;Apc^Min/+^* tumors behave like normal epithelia (*Figure 3G–L*, *Figure 3—figure supplement 1D–I*). Consistently, RNA-seq transcriptome experiments showed that *Ctnnb1^Δi.enh^;Apc^Min/+^* CRCs significantly downregulated genes associated with the Wnt signaling pathway, pluripotency of stem cells, and gastric cancer, whereas increased expression of genes related to normal epithelial functions including oxidative phosphorylation, adhesion, protein digestion and absorption, and mucin biosynthesis (*Figure 3M–O*). Thus, ablation of ieCtnnb1 deters tumorigenesis of intestinal cancers by suppressing the Wnt signaling.

## Human ieCTNNB1 drives the expression of reporter gene in the GI tract

We next investigated whether the human genome also harbors an intestinal enhancer for *CTNNB1*. A region with open chromatin (DNase I hypersensitivity) and enrichment of H3K27ac/H3K4me1 was identified at 38,489 bp upstream of human *CTNNB1* in adult small intestine and colon tissues (*Figure 4A*). Moreover, the H3K27ac enrichment is also present in esophagus but not in non-GI tissues (*Figure 4—figure supplement 1A*). Similar to ieCtnnb1, the 3-kb-long human genomic region, named as ieCTNNB1, resides in the same TAD as the promoter of *CTNNB1* (*Figure 4—figure supplement 1B*). We generated ieCTNNB1 reporter mice (*H11^hi.enh^*) and observed that ieCTNNB1 could drive *LacZ* expression in the bottom of small intestinal crypts and the entire epithelia of large intestine, also exhibiting a proximal-low to distal-high pattern (*Figure 4B*). Notably, the LacZ signal driven by human ieCTNNB1 was less intense than that driven by mouse ieCtnnb1, potentially attributed to species-specific TF-enhancer interactions.

The H3K27ac-enriched region of ieCTNNB1 was divided into five equal parts for the luciferase reporter assay. Experiments demonstrated that each of the five ieCTNNB1 subregions individually increased luciferase activity in both HCT116 and HeLa cells (*Figure 4C*, *Figure 4—figure supplement 1C*). We then conducted CRISPR-mediated activation and interference in HCT-15 CRC cells using three guide RNAs (gRNAs) targeting ieCTNNB1, which respectively enhanced and inhibited transcription of *CTNNB1* (*Figure 4D*). Together, ieCTNNB1 is the intestinal enhancer of *CTNNB1* for human GI epithelia.

## ieCTNNB1 is activated in CRC and its activity positively correlates with the expression of *CTNNB1*

CRC, one of the most prevalent cancers in human, is often associated with aberrant activation of the Wnt signaling (*Li et al., 2021a*). We then studied whether the occurrence of CRC is associated with enhancer activation of ieCTNNB1. We previously conducted paired analyses of chromatin immunoprecipitation sequencing (ChIP-seq) for H3K27ac and H3K4me3, alongside RNA-seq on 68 CRC samples and their adjacent normal (native) tissue (*Li et al., 2021a*). In the current study, we performed analyses for the enrichment of H3K27ac and H3K4me3 at ieCTNNB1 and *CTNNB1* promoter regions, as well as the expression levels of *CTNNB1*, followed by combined analyses (*Figure 5A*, *Figure 5—figure supplement 1*). As anticipated, the expression levels of *CTNNB1* in CRC samples were significantly higher than those in neighboring native tissues (*Figure 5B*). Essentially, the enrichment of H3K27ac at ieCTNNB1 in CRC tissues were significantly higher than that in native tissues (*Figure 5C*, *Figure 5—figure supplement 1A*). In contrast, the enrichment of H3K4me3 at *CTNNB1's* promoter was comparable between cancer and native tissues (*Figure 5C*). Moreover, the enrichment of H3K27ac at ieCTNNB1 positively correlated with the expression of *CTNNB1*, but the enrichment of H3K4me3 at *CTNNB1's* promoter was not associated with *CTNNB1's* expression (*Figure 5D*). Therefore, it's the activity of enhancer but not the promoter that reflects the occurrence of CRC and expression levels of *CTNNB1*.

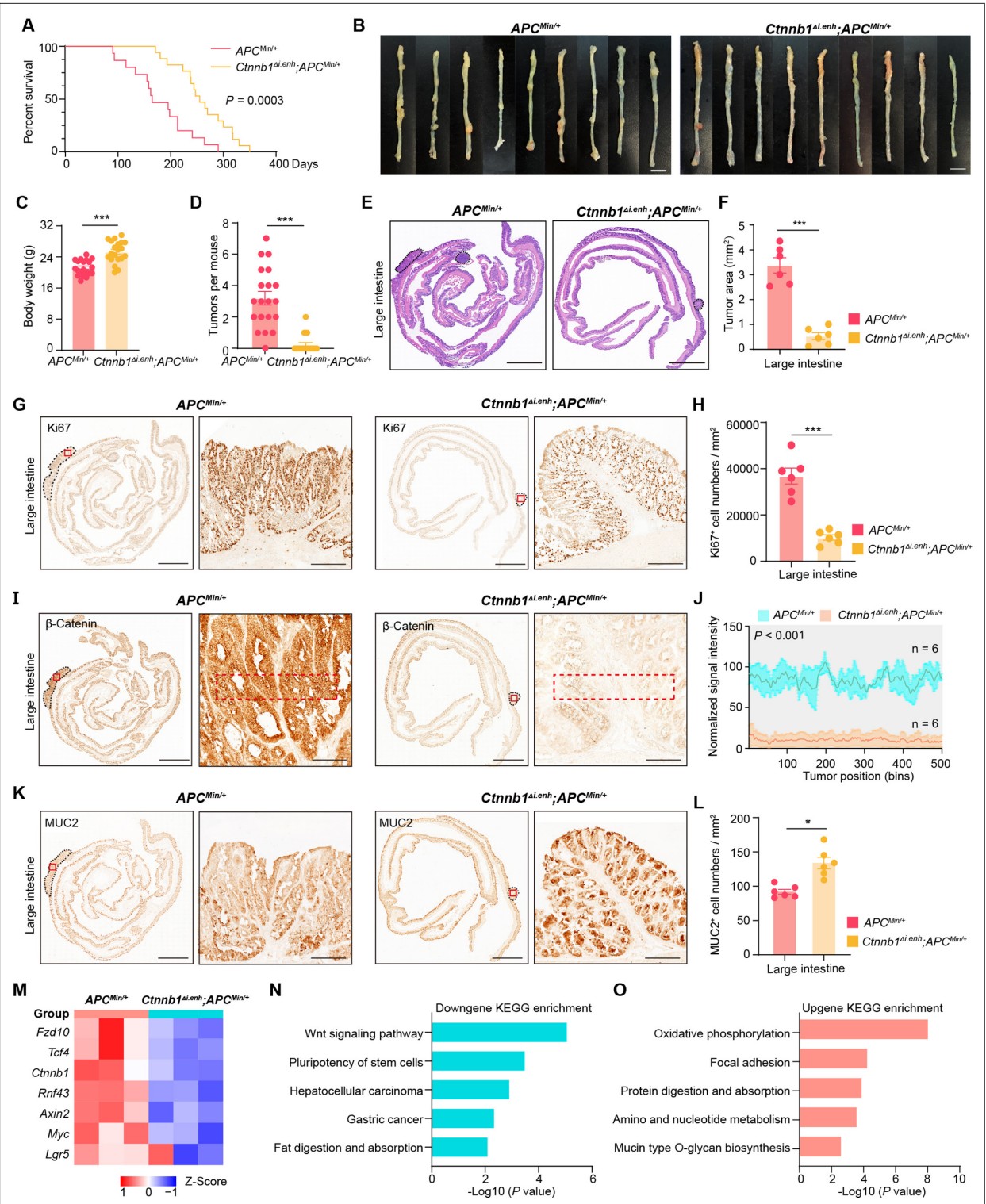

**Figure 3.** Knocking out ieCtnnb1 inhibits tumorigenesis of colorectal cancer. (**A**) Survival of *Apc*^Min/+ (n=15) and *Ctnnb1*^Δi.enh^;*Apc*^Min/+ (n=17) mice. (**B**) Colon images of 5-month-old *Apc*^Min/+ (n=9) and *Ctnnb1*^Δi.enh^;*Apc*^Min/+ (n=9) mice. (**C**) Weight statistics of 5-month-old *Apc*^Min/+ (n=20) and *Ctnnb1*^Δi.enh^;*Apc*^Min/+ (n=20) mice. (**D**) The statistical analysis of tumor counts in colons of 5-month-old *Apc*^Min/+ (n=9) and *Ctnnb1*^Δi.enh^;*Apc*^Min/+ (n=9) mice. (**E**) Representative H&E staining images of colon sections of 5-month-old *Apc*^Min/+ and *Ctnnb1*^Δi.enh^;*Apc*^Min/+ mice. (**F**) The statistical analysis of colon tumor area in 5-month-old *Apc*^Min/+ (n=6) and *Ctnnb1*^Δi.enh^;*Apc*^Min/+ (n=6) mice. (**G–H**) Immunohistochemistry (**G**) and quantification (**H**) of Ki67+ cells in colon tumors of 5-month-old *Apc*^Min/+ (n=6) and *Ctnnb1*^Δi.enh^;*Apc*^Min/+ (n=6) mice. (**I–J**) Immunohistochemistry (**I**) and signal intensity statistics (J, red dashed boxes of I) of β-catenin in colon tumors of 5-month-old *Apc*^Min/+ (n=6) and *Ctnnb1*^Δi.enh^;*Apc*^Min/+ (n=6) mice. (**K–L**) Immunohistochemistry (**K**) and

*Figure 3 continued on next page*

*Figure 3 continued*

quantification (**L**) of MUC2+ cells in colon tumors of 5-month-old *Apc*$^{Min/+}$ (n=6) and *Ctnnb1*$^{\Delta i.enh}$;*Apc*$^{Min/+}$ (n=6) mice. (**M**) The heatmap showing relative expressions of Wnt signaling pathway genes of colon tumors from 5-month-old *Apc*$^{Min/+}$ (n=3) and *Ctnnb1*$^{\Delta i.enh}$;*Apc*$^{Min/+}$ (n=3) mice. (**N–O**) KEGG analyses of downregulated (**N**) and upregulated (**O**) genes in colon tumors of 5-month-old *Apc*$^{Min/+}$ (n=3) and *Ctnnb1*$^{\Delta i.enh}$;*Apc*$^{Min/+}$ (n=3) mice. Scale bars, 1 cm (**B**), 4 mm (**E, G, I, and K**), 200 μm (magnified views in G and K), 100 μm (magnified views in I). Quantification data are shown as means ± SEM, statistical significance was determined using an unpaired two-tailed Student's *t*-test (**C, D, F, H, J, and L**) or log-rank analysis (**A**). *p<0.05, **p<0.01, ***p<0.001, and ****p<0.0001. ns, not significant.

The online version of this article includes the following source data and figure supplement(s) for figure 3:

**Source data 1.** Numerical data for *Figure 3A*.

**Source data 2.** Numerical data for *Figure 3C*.

**Source data 3.** Numerical data for *Figure 3D*.

**Source data 4.** Numerical data for *Figure 3F*.

**Source data 5.** Numerical data for *Figure 3H*.

**Source data 6.** Numerical data for *Figure 3L*.

**Figure supplement 1.** Knocking out ieCtnnb1 inhibits tumorigenesis of small intestine.

**Figure supplement 1—source data 1.** Numerical data for *Figure 3—figure supplement 1A*.

**Figure supplement 1—source data 2.** Numerical data for *Figure 3—figure supplement 1C*.

**Figure supplement 1—source data 3.** Numerical data for *Figure 3—figure supplement 1E*.

**Figure supplement 1—source data 4.** Numerical data for *Figure 3—figure supplement 1I*.

By examining Genotype Tissue Expression (GTEx) eQTL data browser, an SNP, rs15981379 (C>T) within ieCTNNB1 was discovered (*Figure 5A*). Importantly, the C>T variation is inversely correlated with the expression levels of *CTNNB1* in the esophagus and transverse colon (*Figure 5E*). Luciferase reporter assays in HCT116 and HeLa cells revealed that the C>T variation indeed compromised luciferase activity driven by the ieCTNNB1 region 1 (*Figure 5F*).

## HNF4α and p-S133-CREB1 bind to ieCTNNB1 and maintains the expression of *CTNNB1*

Characterizing associated trans-factors of ieCtnnb1 and ieCTNNB1 is essential for revealing related transcriptional mechanisms and holds clinical relevance. By examining Cistrome Data Browser for ChIP-seq data, eight trans-factors were found to associate with ieCTNNB1 in human CRC cells or ieCtnnb1 in mouse GI tissues (*Figure 6A*, *Figure 6—figure supplement 1A*). Moreover, they are relatively highly expressed and/or have positive correlation with *CTNNB1*'s expression in colon cancer samples. We then transduced shRNAs-expressing plasmids into HCT-15 cells to individually downregulate their expressions, revealing that decreasing the expression of *CREB1* (cAMP responsive element binding protein 1) or *HNF4A* (hepatocyte nuclear factor 4 alpha) could compromise *CTNNB1*'s transcription (*Figure 6B*, *Figure 6—figure supplement 1B*). Of note, HNF4α plays crucial roles in the normal development of the colon and the maintenance of epithelial barrier (*Garrison et al., 2006*; *Cattin et al., 2009*). The phosphorylation of CREB1 at serine 133 (p-S133-CREB1) is essential for its role in activation of gene transcription (*Carlezon et al., 2005*). ChIP-qPCR experiments showed that enrichment of p-S133-CREB1 was more prominent than CREB1 at ieCtnnb1 in small intestinal crypts and at ieCTNNB1 in HCT-15 cells (*Figure 6C and D*, *Figure 6—figure supplement 1C*). Notably, HNF4α and p-S133-CREB1 bind to multiple subregions of ieCTNNB1 and ieCtnnb1, as well as the promoters of *CTNNB1* and *Ctnnb1* (*pCTNNB1* and *pCtnnb1*) in HCT-15 cells and small intestinal crypts (*Figure 6C and D*, *Figure 6—figure supplement 1C and D*). By analyzing the GEIPA-COAD (colon adenocarcinoma) and READ (rectum adenocarcinoma) database, expression levels of *CREB1* and *HNF4A* were found to be significantly higher in colon and rectum cancer samples than those of normal tissues (*Figure 6E*). Furthermore, expression levels of *CREB1* and *HNF4A* positively correlated with that of *CTNNB1* (*Figure 6F*). HNF4α and p-S133-CREB1 were significantly enriched at the promoter of *Ctnnb1* in *Apc* mutation induced colon cancer tissues (*Figure 6G and H*). Intriguingly, knockout of ieCtnnb1 greatly diminished the association of HNF4α and p-S133-CREB1 with *Ctnnb1*'s promoter (*Figure 6G and H*). In summary, HNF4α and p-S133-CREB1 were identified as essential trans-factors that bind to ieCTNNB1 and maintain *CTNNB1*'s expression.

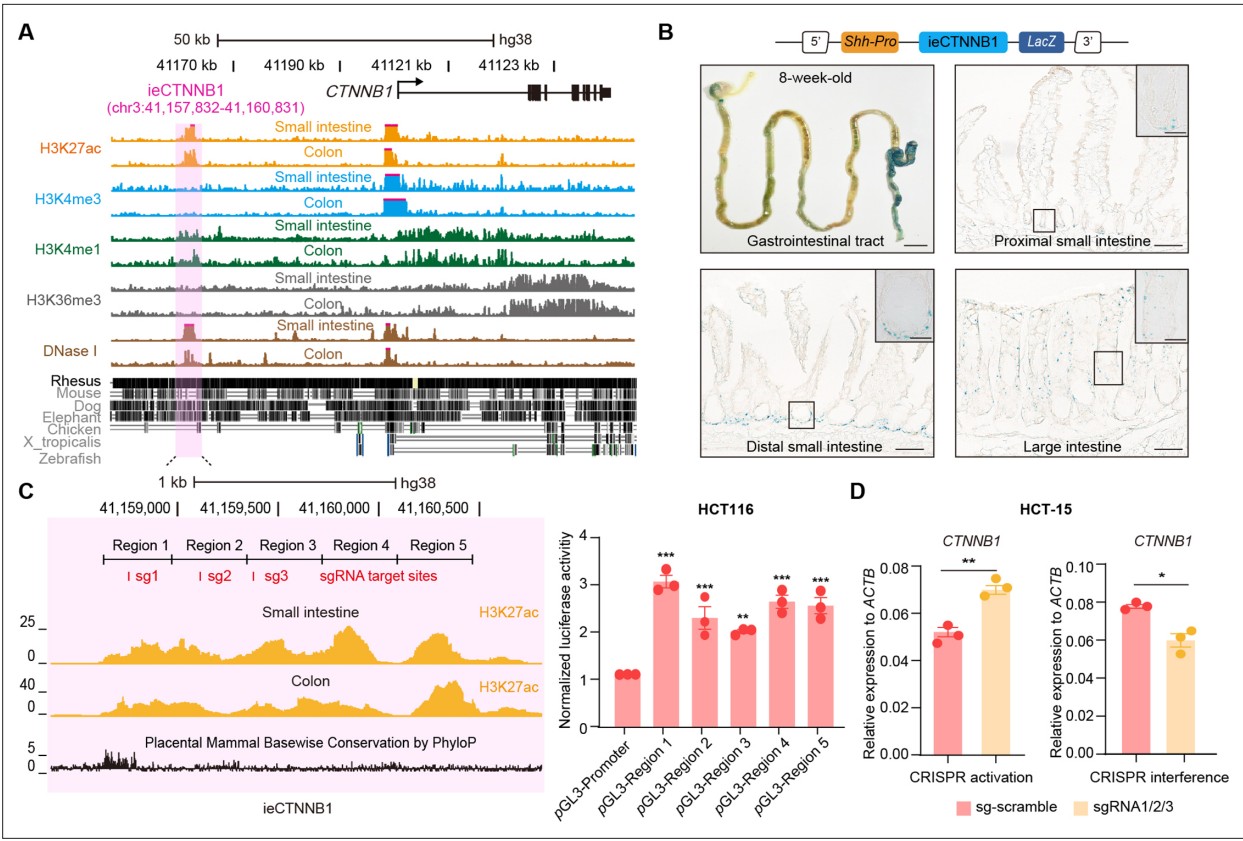

**Figure 4.** ieCTNNB1 is the intestinal enhancer of human *CTNNB1*. (**A**) Schematic representation of human *CTNNB1* gene and the location of ieCTNNB1 (3,000 bp, pink shading), which is marked by H3K27ac and H3K4me1 peaks, and DNaseI hypersensitivity in human small intestine (30-year-old female) and colon (34-year-old male). Data were obtained from ENCODE. (**B**) Top: a schematic illustration showing that the knock-in construct containing the *Shh* promoter, ieCTNNB1 sequences (3,000 bp), and the LacZ reporter gene. Bottom: X-Gal staining (blue) of the gastrointestinal tract, and sections of the proximal small intestine, distal small intestine, and large intestine in 8-week-old *H11$^{hi.enh}$* mice. (**C**) Left: ieCTNNB1 is marked by enrichment of H3K27ac in human small intestine (30-year-old female) and colon (34-year-old male). Data were obtained from ENCODE. Locations of single-guide RNA (sgRNA) target sites were indicated. Five subregions of ieCTNNB1 were shown. Right: luciferase reporter assay in HCT116 cells transfected with indicated plasmids for 48 hr. (**D**) Quantitative reverse transcription PCR (RT-qPCR) showing relative mRNA levels of *CTNNB1* in HCT-15 cells transfected with indicated CRISPR activation or CRISPR interference vectors for 48 hr. Scale bars, 1 cm (whole mount in B), 100 μm (sections in B), 10 μm (magnified views in B). Quantification data are shown as means ± SEM, statistical significance was determined using one-way ANOVA (**C**) and an unpaired two-tailed Student's *t*-test (**D**). *p<0.05, **p<0.01, ***p<0.001, and ****p<0.0001. ns, not significant.

The online version of this article includes the following source data and figure supplement(s) for figure 4:

**Source data 1.** Numerical data for *Figure 4C*.

**Source data 2.** Numerical data for *Figure 4D*.

**Figure supplement 1.** ieCTNNB1 is the intestinal enhancer of human *CTNNB1*.

**Figure supplement 1—source data 1.** Numerical data for *Figure 4—figure supplement 1C*.

## Discussion

The intestinal epithelia replenish themselves rapidly throughout lifetime, which enables regeneration while predisposes to GI cancers (*Clevers, 2013*). The optimal strength and dynamics of canonical Wnt/β-catenin signaling is central to both homeostasis and tumorigenesis of GI tracts (*Pinto et al., 2003*; *Maretto et al., 2003*; *Beumer and Clevers, 2016*). While the stabilization and nuclear translocation of β-catenin is the key step for Wnt signaling activation, the transcriptional control of *Ctnnb1* could be another layer of regulation (*Zhang and Wang, 2020*; *Rim et al., 2022*). Here, we unveiled an enhancer-dependent transcriptional machinery that tunes expressions of *Ctnnb1* in multiple cell types of intestinal crypts, thereby balancing homeostasis and tumorigenesis of intestinal epithelia (*Figure 7*).

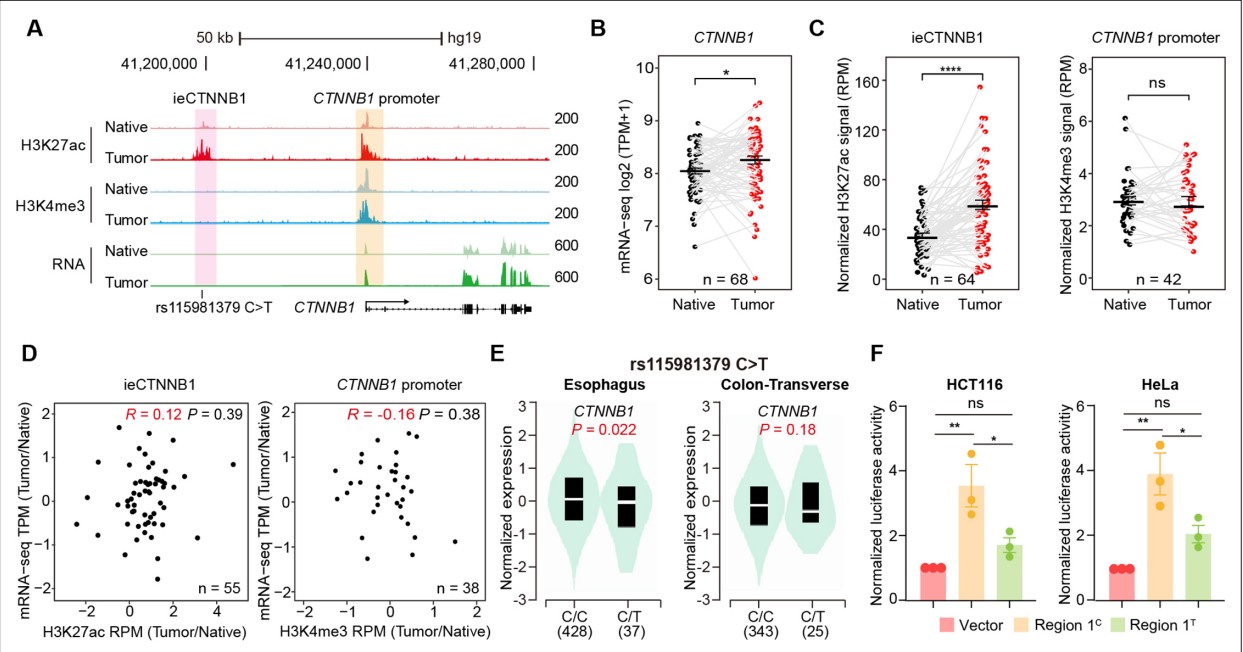

**Figure 5.** ieCTNNB1 is activated in colorectal cancer and its activity positively correlates with the expression of *CTNNB1*. (**A**) Schematic representation of ieCTNNB1 (pink shading) and *CTNNB1* promoter (yellow shading), which is respectively marked by H3K27ac and H3K4me3 peaks, and mRNA signals in native and tumor tissues of a patient with colorectal cancer. The location of risk mutation site was indicated. (**B**) Comparison of *CTNNB1* expression levels in native and tumor tissues of colorectal cancer patients (n=68). (**C**) Left: comparison of H3K27ac signals at ieCTNNB1 in native and tumor tissues of colorectal cancer patients (n=64). Right: comparison of H3K4me3 signals at *CTNNB1* promoter in native and tumor tissues of colorectal cancer patients (n=42). (**D**) Left: correlation between H3K27ac signals at ieCTNNB1 and *CTNNB1* expression in native and tumor tissues of colorectal cancer patients (n=55). Right: correlation between H3K4me3 signals at *CTNNB1* promoter and *CTNNB1* expression in native and tumor tissues of colorectal cancer patients (n=38). (**E**) Left: comparison of *CTNNB1* expression in esophagus between subjects with common sequence (C/C, n=428) and variant sequence (C/T, n=37). Right: comparison of *CTNNB1* expression in transverse colon between subjects with common sequence (C/C, n=343) and variant sequence (C/T, n=25). (**F**) Luciferase reporter assay in HCT116 and HeLa cells transfected with indicated plasmids for 48 hr. Quantification data are shown as means ± SEM, statistical significance was determined using a paired (**B, C, and D**) or unpaired (**E**) two-tailed Student's *t*-test and two-way ANOVA (**F**). *p<0.05, **p<0.01, ***p<0.001, and ****p<0.0001. ns, not significant. R: Pearson correlation.

The online version of this article includes the following source data and figure supplement(s) for figure 5:

**Source data 1.** Numerical data for *Figure 5F*.

**Figure supplement 1.** ieCTNNB1 is activated in colorectal cancer.

Our findings reinforce the notion that the canonical Wnt signaling is required for the establishment of the secretory cell lineage (*Pinto et al., 2003*; *van Es et al., 2005*; *Pinto and Clevers, 2005*). Expression levels of markers for secretory cells were greatly reduced whereas levels of markers for absorptive cells were elevated in all small intestinal crypt cells of *Ctnnb1^{Δi.enh}* mice. Fate determinations of CBC daughter cells occur upon leaving the stem cell niche, exposed to changing WNT, Notch, and EGF levels. Progenitors losing contact with PC Notch ligands upregulate DLL1 and DLL4, commit to the secretory lineage, and induce Notch activity in neighboring cells, promoting EC differentiation (*van Es et al., 2012*; *Beumer and Clevers, 2021*).

WNT signaling not only directs PC-fated progenitors toward the crypt bottom but also directly induces their differentiation (*van Es et al., 2005*; *Andreu et al., 2008*; *Beumer and Clevers, 2021*). Single-cell sequencing and reporter gene analyses indicated that the activity of ieCtnnb1 is present in PCs. Knockout of ieCtnnb1 decreases number of PCs by almost 50% but the pool size of CBCs was not significantly altered. A previous study showed that genetic removal of PCs leads to the concomitant loss of Lgr5+ stem cells (*Sato et al., 2011*), while other studies indicated that PCs are not mandatory niche cells for the intestinal epithelia (*Garabedian et al., 1997*; *Durand et al., 2012*; *Kim et al., 2012*). Although Wnt ligands provided by PCs may support Lgr5+ CBCs, essential Wnt ligands are primarily supplied by subepithelial stromal populations (*Degirmenci et al., 2018*; *Shoshkes-Carmel*

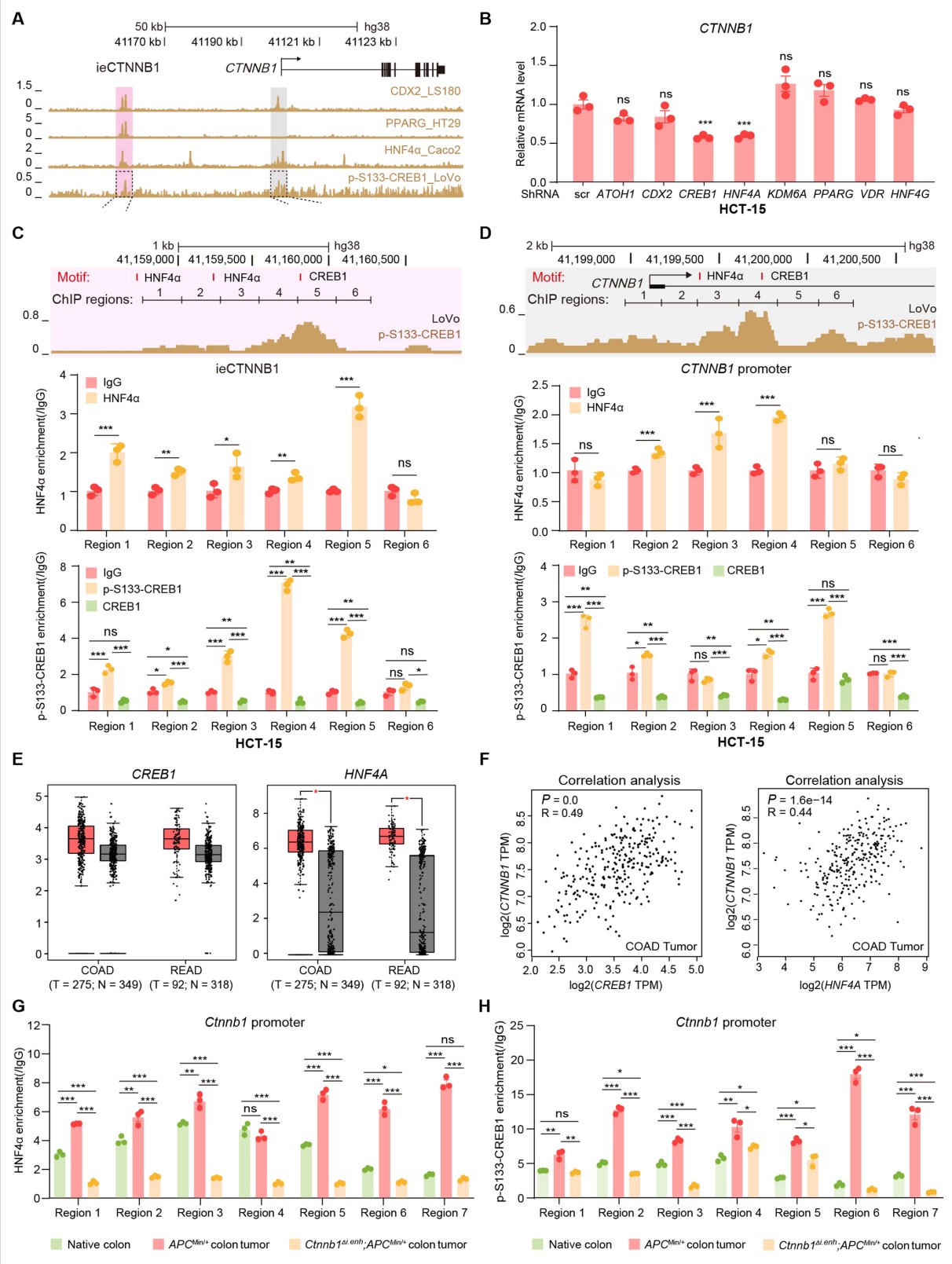

**Figure 6.** HNF4α and p-S133-CREB1 associate with ieCTNNB1 to regulate *CTNNB1*'s transcription. (**A**) Chromatin immunoprecipitation sequencing (ChIP-seq) tracks of indicated trans-acting factors enriched at ieCTNNB1 (pink shading) and *CTNNB1* promoter (gray shading) in indicated colorectal cancer cell lines. Shaded regions were enlarged in C (pink) and D (gray) respectively. (**B**) Quantitative reverse transcription PCR (RT-qPCR) showing relative mRNA levels of *CTNNB1* in HCT-15 cells transfected with indicated shRNA-expressing plasmids for 48 hr. The expression level of *CTNNB1* in

*Figure 6 continued*

cells transfected with scramble (scr) shRNA was set to '1'. (**C–D**) Top: schematic diagram showing the enrichment of p-S133-CREB1 at ieCTNNB1 (**C**) and *CTNNB1*'s promoter (**D**). Locations of HNF4α and CREB1 binding motif sites were indicated. Middle and bottom: ChIP-qPCR showing enrichment of HNF4α (middle), CREB1 and p-S133-CREB1 (bottom) at ieCTNNB1 (**C**), and *CTNNB1*'s promoter (**D**) in HCT-15 cells. Locations of ChIP regions were indicated. (**E**) Comparison of expression levels of *CREB1* and *HNF4A* between native and tumor tissues in colon adenocarcinoma (COAD) and rectum adenocarcinoma (READ) tumors. (**F**) Correlations between the expression level of *CTNNB1* and those of *CREB1* or *HNF4A* in COAD tumors. (**G–H**) ChIP-qPCR showing enrichment of HNF4α (**G**) and p-S133-CREB1 (**H**) at *Ctnnb1* promoter in native colon tissues of WT (n=3) mice, tumor tissues of *Apc^Min/+* (n=3) mice, and *Ctnnb1^Δi.enh;Apc^Min/+* (n=3) mice. Quantification data are shown as means ± SEM, statistical significance was determined using one-way ANOVA (**B**), unpaired two-tailed Student's *t*-test (**E and F**), and Multiple *t*-tests – one per row (**C, D, G, and H**). *p<0.05, **p<0.01, ***p<0.001, and ****p<0.0001. ns, not significant. R: Pearson correlation.

The online version of this article includes the following source data and figure supplement(s) for figure 6:

**Source data 1.** Numerical data for *Figure 6B*.

**Source data 2.** Numerical data for *Figure 6C*.

**Source data 3.** Numerical data for *Figure 6D*.

**Source data 4.** Numerical data for *Figure 6G*.

**Source data 5.** Numerical data for *Figure 6H*.

**Figure supplement 1.** HNF4α and p-S133-CREB1 associate with ieCtnnb1 to regulate *Ctnnb1*'s transcription.

**Figure supplement 1—source data 1.** Numerical data for *Figure 6—figure supplement 1B*.

**Figure supplement 1—source data 2.** Numerical data for *Figure 6—figure supplement 1C*.

**Figure supplement 1—source data 3.** Numerical data for *Figure 6—figure supplement 1D*.

**Figure supplement 1—source data 4.** Numerical data for *Figure 6—figure supplement 1E*.

---

*et al., 2018*). The presence of redundancy at the stem cell niche enhances the accuracy in maintaining intestinal homeostasis.

ieCtnnb1 knockout compromises Wnt signaling dosage, leading to PC loss, dysregulation of genes controlling absorptive and secretory functions of intestinal epithelia, and reduced proliferation of stem cells. These combined effects could account for the lighter body weight observed in adult *Ctnnb1^Δi.enh* mice.

On the other hand, colon and small intestinal tumors were rarely observed in *Ctnnb1^Δi.enh;Apc^Min/+* mice. Among the few *Ctnnb1^Δi.enh* tumors analyzed, they exhibited lower *Ctnnb1* expression, reduced proliferation, and behaved more like normal intestinal epithelia. Different truncations or mutations of *APC* lead to specific tumor types throughout the body in a Wnt/β-catenin dosage-dependent manner. For example, although *Apc^1638N;Ctnnb1^+/-* mice were less prone to GI cancers, *Apc^1638N;Ctnnb1^+/-* female animals tended to grow breast cancers (*Bakker et al., 2013*). Therefore, the intestinal specificity of ieCtnnb1 makes it an ideal target to lower the strength of Wnt signaling in treating CRC without affecting other tissues (*Hamdan and Johnsen, 2019*; *Orouji et al., 2022*). Importantly, ieCTNNB1 displayed higher enhancer activity in most CRC samples collected in the study. Moreover, the SNP rs15981379 (C>T) within ieCTNNB1 is associated with the expression of CTNNB1 in the GI tract. Future population studies could investigate how the enhancer activity of ieCTNNB1 and this particular SNP are associated with CRC susceptibility and prognosis.

We noticed that after knocking out ieCtnnb1, the level of β-catenin in the nuclei of small intestinal crypt cells of *Ctnnb1^Δi.enh* mice decreased more significantly compared to that in the cytoplasm (49.5% vs 29.8%). Although the loss of ieCtnnb1 should not directly lead to reduced nuclear translocation of β-catenin, RNA-seq results showed that the loss of ieCtnnb1 causes a reduction in the expression of *Bambi* (BMP and activin membrane-bound inhibitor), a target gene in the canonical Wnt signaling pathway (*Figure 1M*). BAMBI promotes the binding of Frizzled to Dishevelled, thereby stabilizing β-catenin and facilitating its nuclear translocation (*Lin et al., 2008*; *Liu et al., 2014*; *Mai et al., 2014*; *Zhang et al., 2015*). Thus, it is likely that the decreased level of BAMBI resulting from the loss of ieCtnnb1 further reduced nuclear β-catenin.

Both mouse ieCtnnb1 and human ieCTNNB1 were able to drive reporter gene expression at the base of small intestinal crypts and throughout the epithelium of the large intestine. Very intriguingly, unlike the highly conserved neCtnnb1 sequence among amniotes, mouse ieCtnnb1 and human ieCTNNB1 do not share significant homology in primary sequence (*Wang et al., 2022*). Nonetheless, both ieCtnnb1 and ieCTNNB1 localize at the intergenic region 30–40 kb upstream of *Ctnnb1*'s

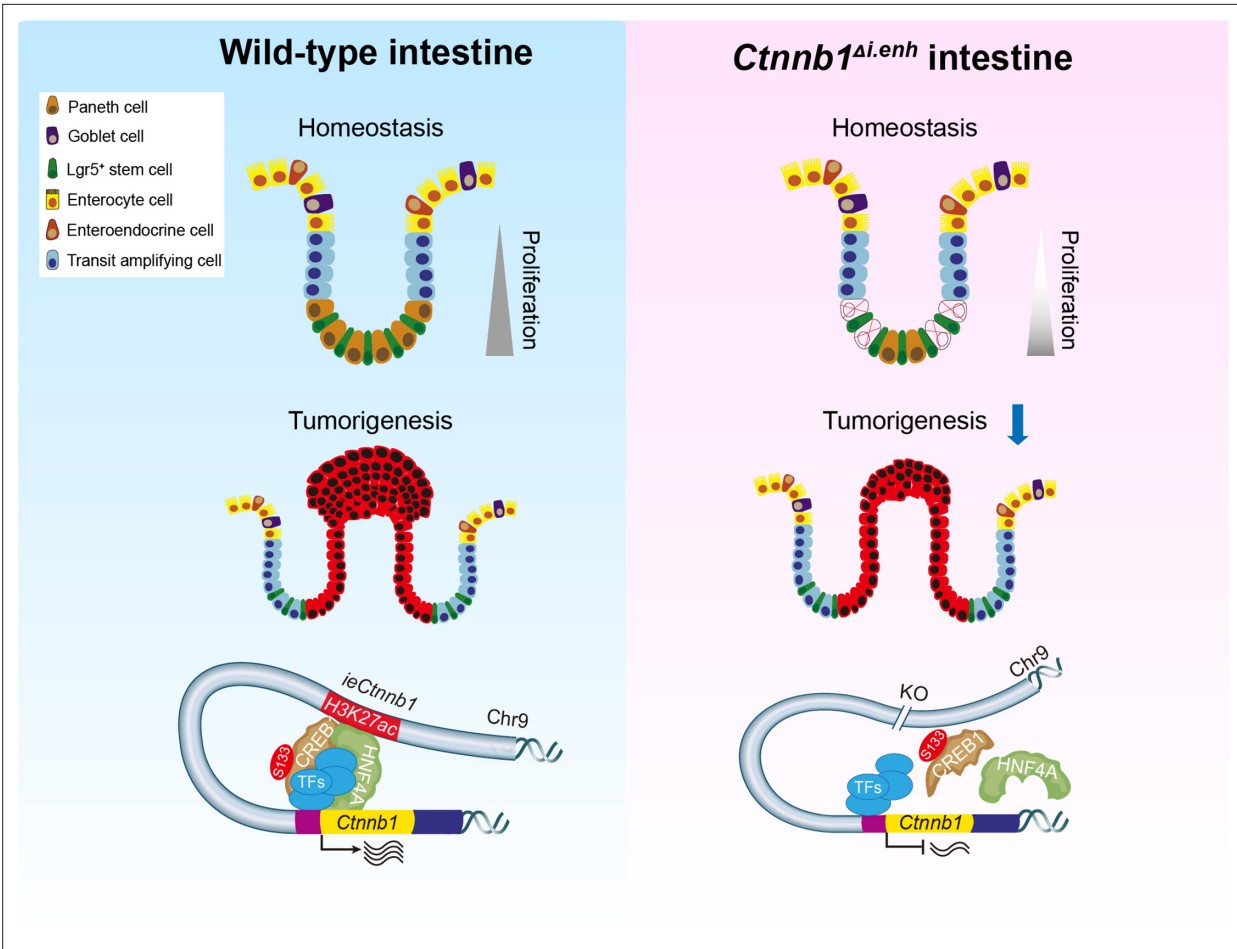

**Figure 7.** The working model. ieCtnnb1, the intestinal enhancer of *Ctnnb1*, balances epithelial homeostasis and tumorigenesis by transcriptionally controlling Wnt signaling dosage.

promoter. Moreover, both ieCtnnb1 and ieCTNNB1 contain binding motifs for HNF4α and CREB1. HNF4α and p-S133-CREB1 were found to associate with ieCtnnb1 and *Ctnnb1*'s promoter to maintain *Ctnnb1*'s expression. These findings support the notion that conserved TF binding motifs stabilize and preserve enhancer functionality over evolution (*Zabidi and Stark, 2016*; *Wong et al., 2020*). Of note, ieCtnnb1 and ieCTNNB1 contain distinct TF binding motifs, which may respectively exert species-specific roles in regulating the transcription of *Ctnnb1*/*CTNNB1* (*Figure 6—figure supplement 1E*). Future studies will dissect common and distinct features of motif composition, location, order, and affinity with trans-factors within ieCtnnb1 and ieCTNNB1. This analysis will provide insights how sequence variations of ieCtnnb1 are adapted to intestinal homeostasis of different species. Finally, it would be intriguing to explore how neocortical and intestinal enhancers of *Ctnnb1* controls tissue- and stage-specific transcription, as well as the existence of other potential enhancers of *Ctnnb1*.

## Methods

### Key resources table

| Reagent type (species) or resource | Designation | Source or reference | Identifiers | Additional information |
|---|---|---|---|---|
| Gene (*Mus musculus*) | *Ctnnb1* | GenBank | Gene ID: 12387 | |
| Gene (*Homo sapiens*) | *CTNNB1* | GenBank | Gene ID: 1499 | |

*Continued on next page*

*Continued*

| Reagent type (species) or resource | Designation | Source or reference | Identifiers | Additional information |
|---|---|---|---|---|
| Genetic reagent (*Mus musculus*) | *ieCtnnb1* knockout | This paper | N/A | Received from Beijing Biocytogen, China |
| Genetic reagent (*Mus musculus*) | *ieCtnnb1-LacZ* | This paper | N/A | Received from Shanghai Model Organisms Center Inc, China |
| Genetic reagent (*Homo sapiens*) | *ieCTNNB1-LacZ* | This paper | N/A | Received from Shanghai Model Organisms Center Inc, China |
| Strain, strain background (*Mus musculus*) | BAT-Gal | The Jackson Laboratory | JAX#005317; RRID:IMSR_JAX:005317 | |
| Strain, strain background (*Mus musculus*) | *Lgr5*-EGFP-IRES-CreERT2 | The Jackson Laboratory | JAX#008875; RRID:IMSR_NM-KI-200154 | |
| Strain, strain background (*Mus musculus*) | *Apc^{Min/+}* | The Jackson Laboratory | JAX#002020; RRID:MGI:3834882 | |
| Cell line (*Homo sapiens*) | HEK293T | ATCC | CRL-1573; RRID:CVCL_0045 | |
| Cell line (*Homo sapiens*) | HCT116 | China Center for Type Culture Collection | GDC0625; RRID:CVCL_0291 | |
| Cell line (*Homo sapiens*) | HCT-15 | China Center for Type Culture Collection | GDC0250; RRID:CVCL_0292 | |
| Cell line (*Homo sapiens*) | HeLa | China Center for Type Culture Collection | GDC0009; RRID:CVCL_0030 | |
| Transfected construct (*Homo sapiens*) | pHR-SFFV-dCas9-BFP-KRAB | Addgene | RRID:Addgene #46911 | |
| Transfected construct (*Homo sapiens*) | lenti sgRNA (MS2)_zeo backbone vector | Addgene | RRID:Addgene #61427 | |
| Transfected construct (*Homo sapiens*) | dCAS9-VP64_GFP | Addgene | RRID:Addgene #61422 | |
| Transfected construct (*Homo sapiens*) | lenti MS2-P65-HSF1_Hygro | Addgene | RRID:Addgene #61426 | |
| Antibody | Anti-α-defensin5 (Rabbit monoclonal) | Abcam | CAT#ab180515; RRID:AB_2923196 | IHC (1:500) |
| Antibody | Anti-Ki67 (Rabbit polyclonal) | Abcam | CAT#ab15580; RRID:AB_443209 | IHC (1:500) |
| Antibody | Anti-β-catenin (Rabbit monoclonal) | Cell Signaling Technology | CAT#8480T; RRID:AB_11127855 | IHC (1:100) WB (1:1000) |
| Antibody | Anti-HNF4α (Rabbit monoclonal) | Abcam | CAT#ab181604; RRID:AB_2890918 | ChIP (1 µg/ml) |
| Antibody | Anti-CREB1(S133) (Rabbit monoclonal) | Abcam | CAT#ab32096; RRID:AB_731734 | ChIP (1 µg/ml) |
| Antibody | Anti-Muc2 (Rabbit polyclonal) | Thermo Fisher | CAT#PA5-21329; RRID:AB_11153058 | IHC (1:500) |
| Antibody | Anti-EpCAM FITC (Mouse monoclonal) | eBioScience | CAT#347197; RRID:AB_400261 | FACS (1:200) |
| Recombinant DNA reagent | pGL3-promoter-ieCtnnb1 | This paper | N/A | Obtained from Zhou Lab of Wuhan University |
| Recombinant DNA reagent | pGL3-promoer-ieCTNNB1 Region1 | This paper | N/A | Obtained from Zhou Lab of Wuhan University |
| Recombinant DNA reagent | pLKO.1-shRNAs | This paper | N/A | Obtained from Zhou Lab of Wuhan University |
| Sequence-based reagent | TF-shRNAs | This paper | PCR primers | See *Supplementary file 1* |

*Continued on next page*

*Continued*

| Reagent type (species) or resource | Designation | Source or reference | Identifiers | Additional information |
|---|---|---|---|---|
| Sequence-based reagent | ieCtnnb1/ ieCTNNB1 ChIP-qPCR primers | This paper | ChIP-qPCR primers | See *Supplementary file 1* |
| Sequence-based reagent | Genotyping primers | This paper | PCR primers | See *Supplementary file 1* |
| Commercial assay or kit | Dual-Luciferase Reporter Assay System | Promega | CAT#E1910 | |
| Commercial assay or kit | Alcian Blue PAS Stain Kit | Abcam | CAT#ab245876 | |
| Commercial assay or kit | BeyoClick EdU Cell Promotion Kit | Beyotime | CAT#C0075S | |
| Software | Fiji/ImageJ | https://fiji.sc | v2.0.0; RRID:SCR_003070 | |
| Software | GraphPad Prism | https://www.graphpad.com | v8.0; RRID:SCR_002798 | |
| Other | DAPI | BD Pharmingen | CAT#564907 | (1 µg/ml) |

## Mice and genotyping

All animal procedures were approved by the Animal Care and Ethical Committee of Medical Research Institute at Wuhan University. C57BL/6 background mice were housed in a temperature- and humidity-controlled environment with 12 hr light/dark cycle and food/water ad libitum. $Ctnnb1^{\Delta i.enh}$ mice were generated in the Beijing Biocytogen. $H11^{i.enh}$ reporter mice were generated in the Shanghai Model Organisms Center Inc. BAT-Gal mice were provided by Dr. Jun-Lei Chang (Jackson Lab, stock number 005317). $Lgr5$-EGFP-IRES-CreERT2 mice were gifts from Dr. Ye-Guang Chen (Jackson Lab, stock number 008875). $APC^{Min/+}$ mice were gifts from Dr. Bo Zhong (Jackson Lab, stock number 002020). The primer set forward 5′-atcctctgcatggtcaggtc-3′/reverse 5′-cgtggcctgattcattcc-3′ was used for genotyping of BAT-Gal mice, $H11^{i.enh}$ mice, and $H11^{hi.enh}$ mice with a band size of 315 base pairs (bp). The primer set forward 5′-gtcctgtccgtcactattatcctggc-3′/reverse 5′-ccactgccctgctaaagcattggt-3′/reverse 5′-tgattagttctccgggaagcccagt-3′ were used for genotyping of $Ctnnb1^{\Delta i.enh}$ mice and band sizes for $Ctnnb1^{\Delta i.enh}$ mice are 482 bp (wild-type allele) and 867 bp (ieCtnnb1 knockout allele). The primer set forward *5′-atctcatggcaaacagacct-3′/reverse 5′-tcacaaatcatctcgcaga-3′* was used for genotyping of $APC^{Min/+}$ mice with a band size of 340 bp. The primer set *forward 5′-ctgctctctgctcccagtct-3′/reverse 5′-ataccccatccctttgagc-3′/reverse 5′-gaacttcagggtcagcttgc-3′* were used for genotyping of $Lgr5$-EGFP-IRES-CreERT2 mice and the band sizes for 386 bp (wild-type allele) and 119 bp (mutant allele).

## Cell lines

HEK293T, HCT116, HCT-15, and HeLa cells were purchased from the China Center for Type Culture Collection. The identity of all cell lines has been authenticated through STR profiling and the mycoplasma contamination testing is negative. HEK293T, HeLa, and HCT116 cells were grown in DMEM (Gibco) supplemented with 10% fetal bovine serum (FBS) (Life Technologies) and 1× penicillin/streptomycin (Gibco). HCT-15 cells were maintained in RPMI-1640 (Gibco) containing 10% FBS (Life Technologies) and 1× penicillin/streptomycin (Gibco). For routine culturing, cells were dissociated using 0.25% trypsin (Gibco).

## Tissue fixation and section

Intestinal and tumor tissues were dissected and cleaned in pre-cooled phosphate buffered saline (PBS) and immersed in 4% paraformaldehyde (PFA) overnight at 4°C. For sectioning, intestines were embedded in Tissue-Tek O.C.T. Compound (SAKURA) and cut to 8 µm thickness with a cryostat (Leica CM1950).

## CRISPRa and CRISPRi assay

CRISPR/dCas9-mediated transcription activation (CRISPRa) and CRISPR/dCas9-mediated transcription interference (CRISPRi) assay were performed as previously described (*Wang et al., 2022*). In brief, single-guide RNAs (sgRNAs) were designed using online tool (https://zlab.bio/guide-design-resources)

and were listed in *Supplementary file 1*. sgRNAs were cloned into lenti sgRNA (MS2)_zeo backbone vector (Addgene, #61427). For CRISPRi assays, pHR-SFFV-dCas9-BFP-KRAB (Addgene, #46911) is used. For CRISPRa assays, lenti MS2-P65-HSF1_Hygro (Addgene, #61426) and dCAS9-VP64_GFP (Addgene, #61422) were used. To obtain lentiviral particles, HEK293T cells ($5 \times 10^6$ cells per 10 cm dish) were transiently transfected with 12 µg CRISPRa or CRISPRi constructs, 6 µg psPAX2, and 6 µg pMD2.G. The supernatant containing lentivirus particles was harvested at 72 hr after transfection and filtered through Millex-GP Filter Unit (0.45 µm pore size, Millipore). To obtain high virus titer, the supernatant was centrifuged at $10,000 \times g$ for 100 min and concentrate to 100 µl. On the day before infection, HCT-15 cells ($2 \times 10^5$ cells per well) were inoculated in 12-well plates. 10 µl high titer viruses were added to each well. 72 hr after infection, cells were harvested for RT-qPCR analysis.

## RNA isolation and cDNA synthesis

RNAiso Plus (TAKARA) is used for RNA isolation. 1 ml RNAiso Plus was added to DNase/RNase free EP tube to lyse tissues or cells, followed by adding one-fifth volume of chloroform to achieve phase separation. After vigorous shaking, centrifuge at $12,000 \times g$ for 15 min at 4°C. The upper aqueous phase was transferred to a new DNase/RNase-free EP tube and equal volume of isopropanol was added to precipitate RNA. Precipitation was dissolved in DNase/RNase-free water. Complementary DNAs (cDNAs) were synthesized by HiScript II Q RT SuperMix for qPCR kit (Vazyme; R222-01).

## Real-time RT-qPCR

cDNA was subjected to RT-qPCR using the SYBR Green assay with 2× SYBR Green qPCR master mix (Bimake). PCRs were performed on a CFX Connect Real-Time PCR Detection System (Bio-Rad) under the following condition: 5 min at 95°C, 40 cycles at 95°C for 15 s, 60°C for 20 s. The relative transcript level of each gene was normalized to *Gapdh* (mice) or *ACTB* (human) and calculated according to the $2^{-\Delta\Delta Ct}$ method. All primers used are shown in *Supplementary file 1*.

## Isolation of murine intestinal crypt cells

Murine small intestines were harvested and cut open longitudinally. After three times wash in cold PBS with penicillin (200 U/ml)+streptomycin (200 µg/ml) (Invitrogen), small intestines were cut into 5 cm segments and incubated in pre-cold PBS with 8 mM EDTA at 4°C for 2 hr. Then intestinal tissues were transferred into a centrifuge tube filled with new cold PBS. Shake vigorously for 5 min and collect intestinal epithelial cells (villi plus crypts) from the supernatant. Centrifuge the supernatant to collect all epithelial cells or pass through 70 µm cell filter (Corning) to filter out villi. Crypts were spun down in a centrifuge at $300 \times g$ for 5 min.

## Enriching *Lgr5*-EGFP⁺ cells

Crypts were collected from *Lgr5*-EGFP and *H11^{i.enh}*;*Lgr5*-EGFP mice, followed by resuspension and single-cell digestion in DMEM containing 0.25% trypsin (Gibco) at 37°C for 5 min. Disaggregated cells were passed through 40 µm cell strainers (Corning). *Lgr5*-EGFP⁺ and *Lgr5*-EGFP⁻ cells were collected using a BD Arial III cell sorter.

## X-Gal staining

X-Gal staining is performed as described (*Wang et al., 2022*). In short, frozen slides were placed in fixed buffer (0.2% PFA, 0.1 M PIPES buffer [pH 6.9], 2 mM $MgCl_2$, 5 mM EGTA) for 10 min, followed by rinse for two times with PBS containing 2 mM $MgCl_2$. Then slides were incubated with detergent solution (0.1 M PBS [pH 7.3], 2 mM $MgCl_2$, 0.01% sodium deoxycholate, 0.02% Nonidet P-40) for 10 min. Slides were incubated with freshly prepared and filtered X-Gal staining solution (0.1 M PBS [pH 7.3], 2 mM $MgCl_2$, 0.01% sodium-deoxycholate, 0.02% Nonidet P-40, 5 mM $K_3Fe(CN)_6$, 5 mM $K_4Fe(CN)_6 \cdot 3H_2O$, and 1 mg/ml X-Gal) overnight in the dark at 37°C. For whole GI tissues, tissues were fixed in freshly prepared 4% PFA for 10 min, followed by rinse with detergent solution for 5 min by three times. Tissues were incubated with freshly prepared and filtered X-Gal staining solution at 37°C for a few hours.

## EdU pulse-labeling experiment

Mice were intraperitoneally injected with EdU (10 mg/kg of body weight). After 4 hr, harvest the intestines for frozen sectioning. Staining was performed using BeyoClick EdU Cell Promotion Kit (C0075S, Beyotime, China) according to the manufacturer's manual.

## ChIP-qPCR assay

Cells were crosslinked with 1% formaldehyde for 10 min at room temperature and then quenched with 0.125 M glycine for 5 min and washed twice with cold PBS. Cells were resuspended with 500 µl lysis buffer (50 mM Tris-HCl, pH 8.0, 0.5% SDS, 5 mM EDTA) and subjected to sonication on ice with an output power of 100 W, 12 min, 0.5 s on, 0.5 s off. After centrifugation, 1% of the supernatant was taken out as input. The DNA fragment lengths were measured by gel electrophoresis, showing fragments between 200 bp and 500 bp, with an average length around 300 bp. The rest of the sonicated lysates was diluted into 0.1% SDS and divided into several equal parts. Immunoprecipitation was performed overnight at 4°C on a rotating wheel with sheared chromatin, protein G agarose beads, and indicated antibodies: 2 µg HNF4α (Abcam, ab181604), 2 µg p-S133-CREB1 (Abcam, ab32096), 2 µg IgG antibody (ABclonal, AC005). The next day, beads were cleaned three times with Wash Buffer I (20 mM Tris-HCl, pH 8.0; 1% Triton X-100; 2 mM EDTA; 150 mM NaCl; 0.1% SDS), Wash Buffer II (20 mM Tris-HCl, pH 8.0; 1% Triton X-100; 2 mM EDTA; 500 mM NaCl; 0.1% SDS), Wash Buffer III (10 mM Tris-HCl, pH 8.0; 1 mM EDTA; 0.25 M LiCl; 1% NP-40; 1% deoxycholate), and TE buffer. Protein-DNA complexes were de-crosslinked in 120 µl elution buffer (0.1 M NaHCO$_3$, 1% SDS, 20 µg/ml proteinase K) and shaken at 65°C overnight. DNA purification kit (TIANGEN) was used to extract DNA fragments for qPCR.

The efficiency of qPCR primers was evaluated using gradient diluted ($10^0$, $10^{-1}$, $10^{-2}$, $10^{-3}$, $10^{-4}$, $10^{-5}$) of mouse or human genomes as templates. After completing qPCR, the dilution factors were plotted on the X-axis using log10 values, and the corresponding Ct value was plotted on the Y-axis to create a standard curve. The equation $y=kx + b$ was derived, and the primer efficiency was calculated using the formula $E=10^{(-1/k)} - 1$. The results indicated that the primer efficiency ranged from 90% to 110%, with no significant differences observed.

## Luciferase reporter assay

The luciferase reporter assay was performed as described previously (*Li et al., 2017*). Briefly, the CR of ieCtnnb1, five regions of ieCTNNB1, and the region containing the mutation site were respectively cloned into pGL3-promoter vector (Promega). HEK293T, HeLa, and HCT116 cells were transfected with pGL3-promoter vector and the plasmids mentioned above. *Renilla* basic vector was co-transfected as a control for normalization of luciferase activity. After 48 hr of transfection, Promega Dual Glo test kit (Promega, E1910) was used to measure luciferase activity according to the manufacturer's instructions.

## IHC staining

Frozen sections were dried at 37°C, fixed with 4% PFA for 10 min, and then washed three times with PBS. Antigen retrieval was performed using citrate buffer (pH 6, epitope retrieval solution) at 95°C for 20 min. Slides were then placed in 3% H$_2$O$_2$ solution for a few hours to remove endogenous catalase. After washing with PBS three times, slides were blocked with 10% normal goat serum for 1 hr, then incubated with the following primary antibodies overnight at 4°C: rabbit anti-α-defensin5 (Abcam, ab80515, 1:500), rabbit anti-Ki67 (Abcam, ab15580, 1:500), rabbit anti-β-catenin (Cell Signaling Technology, 8480T, 1:100), rabbit anti-Muc2 (Thermo Fisher Scientific, PA5-21329, 1:500). The next day, slides were incubated with biotin-labeled secondary antibodies at room temperature for 1 hr. Then, slides were incubated with the avidin-biotin-peroxidase complex (VECTASTAIN Elite ABC system, Vector Labs). Finally, colorization was performed in the reaction solution (Tris-HCl [pH 7.2], 5 mg/ml 3,3'-diaminobenzidine, 0.075% H$_2$O$_2$).

## H&E staining

H&E staining kit (Beyotime Biotech, C0105S) was used for experiments. Frozen slides were dried at 65°C, then dripped with hematoxylin staining solution for 5 min. Excess staining solution was rinsed off with water. Slides were immersed in distilled water for 5 min, then rinsed with 95% ethanol, and

stained with eosin solution for 30 s. Subsequently, slides were put into anhydrous ethanol for 2 min, and cleared with xylene for 5 min. After air dry, slides were sealed with neutral resin.

## Immunoblotting analysis

Crypt cells were lysed by 1× SDS loading buffer at 95°C for 10 min. After short centrifugation, the supernatants were loaded onto 10% SDS-polyacrylamide gel electrophoresis gels. Then, proteins were transferred onto polyvinylidene fluoride membrane (Millipore). Membranes were blocked with 10% non-fat milk in Tris-buffered saline containing 0.3% Tween 20 (TBST; pH 7.4) for 1 hr at room temperature. Immunoblotting was carried out us primary antibody: rabbit anti-β-catenin (Cell Signaling Technology, 8480T, 1:100); mouse anti-GAPDH (ABclonal, AC001, 1:1000); rabbit anti-H3 (ABclonal, A2348, 1:10,000). After three times of washing with TBST, the membrane was incubated with anti-rabbit immunoglobulin G (IgG)-conjugated horseradish peroxidase secondary for 1 hr at room temperature. Signals were detected using the ECL substrate (Thermo Fisher Scientific).

## Alcian Blue PAS staining

Alcian Blue PAS Stain Kit (Abcam, ab245876) was used for experiments. All materials and prepared reagents were equilibrated to room temperature and gently agitated prior to use. Tissue sections were applied with acetic acid solution (3%) for 2 min. Then Alcian Blue solution (pH 2.5) was applied for 15–20 min. Slides were rinsed with running tap water for 2 min followed by two changes of distilled water. Periodic acid solution was applied to tissue sections for 5 min. After slides were rinsed with two changes of distilled water, Schiff's solution was applied to tissue sections for 10–20 min. Slides were rinsed with warm running tap water for 2 min followed by two changes of distilled water. Tissue sections were applied with hematoxylin for 2 min. Sections were rinsed in running tap water for 2 min followed by two changes of distilled water. Slides were dehydrated through graded alcohols and sealed with synthetic resin.

## Cell sorting

Crypt single cells were prepared as described above. Enrichment of epithelial cells by cell sorting was carried out according to reported protocols with slight modification (*Haber et al., 2017*; *Huang et al., 2022*). Cells were resuspended using 1 ml PBS and stained with 5 μl anti-EpCAM-FITC (eBioScience, 11-5791-82) at 4°C for 30 min. After washing with PBS three times, cells were resuspended using 1 M FACS buffer (PBS with 1% BSA) containing 0.2 μg/ml DAPI (BD Pharmingen, Cat. No. 564907). Cells were filtered with nylon membrane with a 40 μm pore size. EpCAM positive cells were sorted and collected based on fluorescence signals for subsequent single-cell sequencing. Flow cytometry data were acquired on a BD Arial III flow cytometer and analyzed with FlowJo (version 10.6.2).

## Bulk RNA-seq

RNA-seq and data analyses were performed at Novogene Co., Ltd (Beijing, China). Crypt cells or tumor tissues were harvested and total RNAs were extracted. RNA integrity was assessed using the RNA Nano 6000 Assay Kit of the Bioanalyzer 2100 system (Agilent Technologies, CA, USA). RNA-seq library construction was performed as described previously (*Xu et al., 2021*). Briefly, first strand cDNA was synthesized using random hexamer primer and M-MuLV Reverse Transcriptase (RNase H-). Second strand cDNA synthesis was subsequently performed using DNA Polymerase I and RNase H. Then, the library fragments were purified with AMPure XP system (Beckman Coulter, Beverly, MA, USA). Then PCR was performed with Phusion High-Fidelity DNA polymerase, Universal PCR primers, and Index (X) Primer. At last, PCR products were purified (AMPure XP system). The clustering of the index-coded samples was performed on a cBot Cluster Generation System using TruSeq PE Cluster Kit v3-cBot-HS (Illumina) according to the manufacturer's instructions. After cluster generation, the library preparations were sequenced on an Illumina Novaseq platform and 150 bp paired-end reads were generated.

## Bulk RNA-seq data processing

For data analysis, raw data of fastq format were first processed through in-house perl scripts. In this step, clean data were obtained by removing reads containing adapter, reads containing poly-N (indicating bases that were not detected and hence signal-free), and low-quality reads from the raw data.

Index of the reference genome was built using Hisat2 v2.0.5 and paired-end clean reads were aligned to the reference genome using Hisat2 v2.0.5. featureCounts v1.5.0-p3 was used to count reads numbers mapped to each gene. Differential expression analysis of two groups was performed using the DESeq2 R package (1.20.0). The resulting p-values were adjusted using the Benjamini and Hochberg's approach for controlling the false discovery rate. Genes with an adjusted p-value<0.05 found by DESeq2 were assigned as differentially expressed. GO enrichment analysis of differentially expressed genes was implemented by the clusterProfiler R package, in which gene length bias was corrected. clusterProfiler R package was also used to test the statistical enrichment of differential expression genes in KEGG pathways. Local version of the GSEA tool (http://www.broadinstitute.org/gsea/index.jsp), GO, KEGG, Reactome, DO, and DisGeNET datasets were used for GSEA independently.

## Single-cell RNA-seq and data analysis

Single-cell RNA-seq (scRNA-seq) and data analysis were performed at Novogene Co., Ltd (Beijing, China). The protoplast suspension was loaded into Chromium microfluidic chips with 30 chemistry (v2 or v3) and barcoded with a 10× Chromium Controller (10X Genomics). RNA from the barcoded cells was subsequently reverse-transcribed and sequencing libraries constructed with reagents from a Chromium Single Cell 30 v2 reagent kit (10X Genomics) according to the manufacturer's instructions. Sequencing was performed with Illumina HiSeq 2000 according to the manufacturer's instructions. For data analysis, we use FastQC to perform basic statistics on the quality of the raw reads. Raw reads were demultiplexed and mapped to the reference genome by 10X Genomics Cell Ranger pipeline using default parameters. All downstream single-cell analyses were performed using CellRanger and Seurat unless mentioned specifically. In brief, for each gene and each cell barcode (filtered by Cell-Ranger), unique molecule identifiers were counted to construct digital expression matrices. Secondary filtration by Seurat: a gene with expression in more than three cells was considered as expressed, and each cell was required to have at least 200 expressed genes.

## Hi-C data analysis

Hi-C data analysis completed by Frasergen Medicine Co., Ltd (Wuhan, China). Hi-C data of BALB/c mice large intestinal epithelial cells were provided by Dr. Gen Zheng (*Zheng et al., 2023*). The Hi-C data of SW480 cells was obtained from deposited data (*Orouji et al., 2022*). According to the reported method, the dynamic changes of TAD structures can be explored by calculating the relative signal values of TAD boundaries. TAD boundaries that are common among individuals and are not within the 2 Mb range of chromosomal ends were included, followed by filtering out boundaries with interactions less than 10 in the 2 Mb upstream and downstream regions. The ratio of the interaction between the boundary and its 2 Mb upstream and downstream regions were calculated using log2 to obtain the signal value of the TAD boundary, then the minimum value was subtracted from it to finally obtain the distribution of the relative signal value of each sample TAD boundary.

## ChIP-seq data analysis

The human CRC patient RNA-seq FPKM files and ChIP-seq data were from GEO dataset GSE156614 (*Li et al., 2021a*). All ChIP-seq raw fastq data were removed of adaptor sequence. Cutadapt (version 1.16) was used with the parameters -u 3 -u -10 -U 3 -U -10 -m 30. Cleaned reads were aligned to the human reference genome (hg19) using bowtie2 (version 2.4.4) with default settings. The CrossMap (0.6.5) package was used as a conversion between different versions of the human genome (hg19 and hg38). We calculated the normalized RPM (reads per million mapped reads) of ChIP-seq signal at ieCTNNB1 and *CTNNB1* promoter. Briefly, reads aligned to the specified region were calculated with bedtools multicov command. These reads were normalized with RPM. RNA-seq data FPKM (reads per kilobase per million mapped reads) were converted to TPM (transcript per kilobase per million mapped reads) using R version 4.2.3.

## Knockdown experiments

Target sequences for shRNAs were designed on the Merck Life Sciences website (https://www.sigmaaldrich.cn) and were listed in *Supplementary file 1*. Oligonucleotides for shRNAs were cloned into pLKO.1 vector (Addgene, #21297). The day before transfection, HEK293T cells ($2×10^5$ cells per well) were seeded in 12-well plates. pLKO.1 constructs, psPAX2, and pMD2.G were co-transfected

into HEK293T cells. After 72 hr, supernatant containing virus particles was harvested and passed through Millex-GP Filter Unit (0.45 µm pore size, Millipore). Viral particles were aliquoted and stored at –80°C. On the day before infection, HCT-15 cells ($2 \times 10^5$ cells per well) were seeded in 12-well plates. 20 µl virus suspension were added to each well. After 72 hr, cells were harvested for RT-qPCR to evaluate knockdown efficiency.

## Reference database

ChIP-seq signals for H3K27ac, H3K4me3, H3K4me1, H3K36me3, and DNase I hypersensitivity data of stomach, intestines, liver, and other tissues at different stages were downloaded from the ENCODE (https://www.encodeproject.org/) and their identifiers are listed in *Supplementary file 2*. In addition, ChIP-seq data for listed TFs of cells or tissues were obtained from Cistrome Data Browser (http://cistrome.org). Expressions of *HNF4A* and *CREB1* in human colon (COAD) and rectal (READ) cancers were obtained from GEPIA (gepia.cancer-pku.cn).

## eQTL analysis

The GTEX database (https://www.gtexportal.org/home/) was used to search SNPs at *CTNNB1* and the 1 MB upstream and downstream region, followed by eQTL analysis. A variant, rs115981379 (C>T), was identified inside *CTNNB1*. The expression levels of *CTNNB1* in 428 C/C type and 37 C/T type esophagus samples, and 343 C/C type and 25 C/T type transverse colon samples were also obtained in the GTEX database.

## Analyses of HNF4α and CREB1 binding motifs

The JASPAR database (https://jaspar.elixir.no/) was used to analyze binding motifs of HNF4α and CREB1 at ieCtnnb1/ieCTNNB1 and *Ctnnb1/CTNNB1* promoter. Results are listed in *Supplementary file 3*.

## Nuclear-cytoplasmic separation

Nuclear and cytoplasmic protein extraction kit (Beyotime Biotech, P0027) was used for experiments. Small intestinal crypts were collected from wild-type and $Ctnnb1^{\Delta i.enh}$ mice as described above. According to the manufacturer's instructions, 20 µl cell precipitation was resuspended in 200 µl cytoplasmic protein extraction reagent A with 1 mM PMSF. After placing on ice for 15 min, 10 µl cytoplasmic protein extraction reagent B was added. After vigorous vortex, solutions were centrifuged at $12{,}000 \times g$ for 5 min at 4°C. The supernatant was collected as the extracted cytoplasmic protein. The precipitate was resuspended with 50 µl nuclear protein extraction reagent with 1 mM PMSF. Solutions were placed on ice and vortex vigorously every 2 min for a total of 30 min. After centrifuging at $12{,}000 \times g$ for 10 min at 4°C, the supernatant was collected as the extracted nuclear protein.

## Statistics

All experiments reported in this study were repeated at least three independent times. In vivo analyses were conducted with at least three animals per condition. The number of animals in each experiment is depicted in Figure legends. The signal intensity of tumor stain region was generated using ImageJ software (version 6.0.0.260) as described previously (*Wang et al., 2021*). GraphPad Prism (version 8.0.2) was used to determine statistical significance. Unpaired two-tailed Student's *t*-test were used for analysis between two groups of equal variances by *F*-tests. When the *F*-test of equal variance failed, the Welch *t*-tests were used. Data comparison of three or more groups with control groups were analyzed using one-way ANOVA followed by Dunnett's multiple comparison test. Data comparison of three or more groups among each other were analyzed using one-way ANOVA followed by Tukey's multiple comparison test. The exact p-values were reported in each figure or indicated as ****, $p < 0.0001$; ***, $p < 0.001$; **, $p < 0.01$; *, $p < 0.05$, ns = not significant.

## Acknowledgements

We thank Dr. Yeguang Chen for providing the *Lgr5*-EGFP-IRES-CreERT2 mice. We thank the Core Facility and the Animal Facility of Medical Research Institute of Wuhan University for technical support. We thank all Zhou lab members for critical reading of the manuscript. Y Zhou was supported by grants from National Key R&D Program of China (2022YFA0806603 and 2018YFA0800700), National

Natural Science Foundation of China (32270876 and 31970770), and the Fundamental Research Funds for the Central Universities (2042022dx0003 and 2042023kf0234). Y Liu was supported by grants from National Natural Science Foundation of China (32470776 and 31970770), National Key R&D Program of China (2018YFA0800700 and 2022YFA0806603), and Hubei Natural Science Foundation (2022CFB128).

## Additional information

### Funding

| Funder | Grant reference number | Author |
|---|---|---|
| National Key Research and Development Program of China | 2022YFA0806603 | Yan Zhou |
| National Natural Science Foundation of China | 32270876 | Yan Zhou |
| National Key Research and Development Program of China | 2018YFA0800700 | Yan Zhou |
| National Natural Science Foundation of China | 31970770 | Yan Zhou |
| National Natural Science Foundation of China | 32470776 | Ying Liu |
| Fundamental Research Funds for the Central Universities | 2042022dx0003 | Yan Zhou |
| Fundamental Research Funds for the Central Universities | 2042023kf0234 | Yan Zhou |
| Hubei Natural Science Foundation | 2022CFB128 | Ying Liu |

The funders had no role in study design, data collection and interpretation, or the decision to submit the work for publication.

### Author contributions

Xiaojiao Hua, Investigation, Visualization, Methodology, Writing – original draft, Performed experiments and analyzed data; Chen Zhao, Data curation, Formal analysis, Processed ChIP-seq data; Jianbo Tian, Methodology, Retrieved and analyzed eQTL data; Junbao Wang, Investigation, Methodology, Performed experiments and analyzed data; Xiaoping Miao, Methodology, Retrieved and analyzed eQTL data; Gen Zheng, Resources, Provided Hi-C data; Min Wu, Resources, Provided RNA-seq and ChIP-seq data of colorectal cancer samples; Mei Ye, Resources, Provided RNA-seq and ChIP-seq data of colorectal cancer samples; Ying Liu, Supervision, Funding acquisition, Provided funding and supervised the study; Yan Zhou, Conceptualization, Supervision, Writing – original draft, Project administration, Conceived and designed the study, provided funding and supervised the study and wrote the paper

### Author ORCIDs

Xiaojiao Hua http://orcid.org/0009-0002-3005-9006
Jianbo Tian https://orcid.org/0000-0001-9493-694X
Yan Zhou https://orcid.org/0000-0002-2713-4442

### Ethics

All animal procedures were approved by the Animal Care and Ethical Committee of Medical Research Institute at Wuhan University (Permit Number: MRI2019-LAC46, MRI2021-LAC27 and MRI2022-LAC151). All surgery was performed under sodium pentobarbital anesthesia, and every effort was made to minimize suffering.

Reviewer #1 (Public review): https://doi.org/10.7554/eLife.98238.3.sa1
Reviewer #2 (Public review): https://doi.org/10.7554/eLife.98238.3.sa2
Reviewer #3 (Public review): https://doi.org/10.7554/eLife.98238.3.sa3
Author response https://doi.org/10.7554/eLife.98238.3.sa4

## Additional files

### Supplementary files
- Supplementary file 1. Primers used in this study.
- Supplementary file 2. Summary of high-throughput data used in this study.
- Supplementary file 3. HNF4α and CREB1 motifs analyses at ieCtnnb1/ieCTNNB1 and *Ctnnb1/CTNNB1* promoter.
- MDAR checklist

### Data availability
The GEO accession number for the bulk RNA-seq and scRNA-seq data reported in this paper is GSE233979. Custom codes are described in detail at methods section.

The following dataset was generated:

| Author(s) | Year | Dataset title | Dataset URL | Database and Identifier |
|---|---|---|---|---|
| Hua X, Zhao C, Tian J, Wang J, Miao X, Zheng G, Wu M, Ye M, Liu Y, Zhou Y | 2024 | Wnt signaling dosage controlled by a Ctnnb1 enhancer balances homeostasis and tumorigenesis of intestinal epithelia | https://www.ncbi.nlm.nih.gov/geo/query/acc.cgi?acc=GSE233979 | NCBI Gene Expression Omnibus, GSE233979 |

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
