## [Editor Report · eLife assessment]

Ctnnb1 encodes β-catenin, an essential component of the canonical Wnt signaling pathway. In this **important** study, the authors identify an upstream enhancer of Ctnnb1 responsible for the specific expression level of β-catenin in the gastrointestinal track. Deletion of this enhancer in mice and analyses of its association with human colorectal tumors provide **compelling** support that it controls the dosage of Wnt signaling critical to the homeostasis in intestinal epithelia and colorectal cancers.

---

## [Referee Report · Reviewer #1 (Public review)]

Summary:

Ctnnb1 encodes β-catenin, an essential component of the canonical Wnt signaling pathway. In this study, the authors identify an upstream enhancer of Ctnnb1 responsible for the specific expression level of β-catenin in the gastrointestinal track. Deletion of this promoter in mice and analyses of its association with human colorectal tumors support that it controls the dosage of Wnt signaling critical to the homeostasis in intestinal epithelia and colorectal cancers.

Strengths:

This study has provided convincing evidence to demonstrate the functions of a gastrointestinal enhancer of Ctnnb1 using combined approaches of bioinformatics, genomics, in vitro cell culture models, mouse genetics, and human genetics. The results support the idea that the dosage of Wnt/β-catenin signaling plays an important role in pathophysiological functions of intestinal epithelia. The experimental designs are solid and the data presented are of high quality. This study significantly contributes to the research fields of Wnt signaling, tissue-specific enhancers, and intestinal homeostasis.

Weaknesses:

Insufficient discussion on some findings was a major weakness in the previous submission, which has been addressed in the revised submission.

---

## [Referee Report · Reviewer #2 (Public review)]

Wnt signaling is the name given to a cell-communication mechanism that cells employ to inform on each other's position and identity during development. In cells that receive the Wnt signal from the extracellular environment, intracellular changes are triggered that cause the stabilization and nuclear translocation of β-catenin, a protein that can turn on groups of genes referred to as Wnt targets. Typically these are genes involved in cell proliferation. Genetic mutations that affect Wnt signaling components can therefore affect tissue expansion. Loss of function of APC is a drastic example: APC is part of the β-catenin destruction complex, and in its absence, β-catenin protein is not degraded and constitutively turns on proliferation genes, causing cancers in the colon and rectum. And here lies the importance of the finding: β-catenin has for long been considered to be regulated almost exclusively by tuning its protein turnover. In this article, a new aspect is revealed: Ctnnb1, the gene encoding for β-catenin, possesses tissue-specific regulation with transcriptional enhancers in its vicinity that drive its upregulation in intestinal stem cells. The observation that there is more active β-catenin in colorectal tumors not only because the broken APC cannot degrade it, but also because transcription of the Ctnnb1 gene occurs at higher rates, is novel and potentially game-changing. As genomic regulatory regions can be targeted, one could now envision that mutational approaches aimed at dampening Ctnnb1 transcription could be a viable additional strategy to treat Wnt-driven tumors.

---

## [Referee Report · Reviewer #3 (Public review)]

The authors of this paper identify an enhancer that upstream of the Ctnnb1 gene that selectively enhances expression in intestinal cells. This enhancer sequence drives expression of a reporter gene in the intestine and knockout of this enhancer attenuates Ctnnb1 expression in the intestine, while protecting mice from intestinal cancers. The human counterpart of this enhancer sequence is functional and involved in tumorigenesis. Overall, this is an excellent example of how to fully characterize a cell-specific enhancer. The strength of the study is the thorough nature of the analysis and the relevance of the data to development of intestinal tumors in both mice and humans. A minor weakness was that that loss of this enhancer does not completely compromise expression of Ctnnb1 gene in the intestine, suggesting that other elements are likely involved. The authors have now addressed this concern.

---

## [Author Response]

The following is the authors’ response to the original reviews.

**Reviewer #1:**
(1) One issue that needs to be considered is the nomenclature of the enhancer. The authors have presented data to show this enhancer controls the expression of Ctnnb1 in the stomach, intestine, and colon tissues. However, the name proposed by the authors, ieCtnnb1 (intestinal enhancer of Ctnnb1), doesn't represent its functions. It might be more appropriate to call it a different name, such as gieCtnnb1 (gastrointestinal enhancer of Ctnnb1).

We thank the reviewer for the insightful suggestion and agree that wholemount reporter assays indicated ieCtnnb1 and ieCTNNB1 indeed display activity in the stomach. However, in current study, we focused on the cellular distribution and the function in intestinal epithelia. After careful consideration, we reasoned that the current designation, ieCtnnb1, would be more appropriately represent its expression pattern and functions based on provided evidence. We hope the reviewer could understand our reasoning.

(2) The writing of this manuscript can be improved in a few places.a) The definitions or full names for the abbreviations of some terms, e.g., Ctnnb1, ieCtnnb1, in both abstract and main text, are needed when they first appear. Specifically, Line 108 should be moved to Lines 26 and 95. Lines 125126 are redundant. ieCtnnb1 in Line 130 needs to be defined.

We appreciate the suggestion. In the revision, we have included the definition of Ctnnb1 and the full name of ieCtnnb1 when they first appear in the abstract and the main text. Lines 125-126 were deleted in the revision.

b) Line 192-194, the description of the result needs to be rewritten to reflect

the higher expression of LacZ transcript in eGFP+ cells.

We would like to emphasize that the key point of this part is that the enhancer activity of ieCtnnb1 is present in both Lgr5-eGFP+ and Lgr5-eGFP- cells. This was validated by single-cell sequencing, which revealed the presence of LacZ transcripts in the Paneth cells. Moreover, we could not confidently conclude that eGFP+ cells have higher expression levels of LacZ, as these measurements were obtained from separate, semi-quantitative RT-qPCR experiments.

c) More details are needed for how the data using human tumor samples were generated and how they were analyzed.

We thank the suggestion. In the revision, we have provided additional details regarding the data and subsequent analyses of human CRC samples as follows: ‘We previously conducted paired analyses of chromatin immunoprecipitation sequencing (ChIP-seq) for H3K27ac and H3K4me3, alongside RNA-seq on 68 CRC samples and their adjacent normal (native) tissue (Li et al., 2021). In the current study, we performed analyses for the enrichment of H3K27ac and H3K4me3 at ieCTNNB1 and CTNNB1 promoter regions, as well as the expression levels of CTNNB1, followed by combined analyses (Figure. 5A, Figure 5 - figure supplement 1).’

d) The genomic structures from multiple species are presented at the bottom of Figure 1a. However, the description and explanation are lacking in both the main text and the figure legend.

We apologize for not presenting clearly. We have added related description in the legend of Figure 1A as ‘The sequence conservation of the indicated species is shown at the bottom as vertical lines’. We also added an explanation in lines 162-163 of the main text: ‘Notably, unlike neCtnnb1, the primary sequence of ieCtnnb1 is not conserved among vertebrates (Figure 1A, bottom)’.

**Reviewer #2:**
(1) One of the main issues emerging during reading concerns the interpretation of the consequence of deleting the ieCtnnb1 enhancer. The authors write on line 235 that the deletion of ieCtnnb1 ‘undermined’ Wnt signaling in the intestinal epithelium. This feels too strong, as the status of the pathway is only mildly affected, testified by the observation that mice with homozygous deletion on ieCtnnb1 are alive and well. The enhancer likely ‘only’ drives higher Ctnnb1 expression, and it does not affect Wnt signaling by other mechanisms. The reduction of Wnt target gene expression upon its deletion is easily interpreted as the consequence of reduced β-catenin. Also the title, in my opinion, allows this ambiguity to stick in readers' minds. In other words, the authors present no evidence that the ieCtnnb1 enhancer controls Wnt signaling dosage via any mechanism other than its upregulation of Ctnnb1 expression in the intestinal epithelium. Reduced Ctnnb1, in turn, could explain the observed reduction of Wnt signaling output and the interesting downstream physiological consequences. Unless the authors think otherwise, I suggest they clarify this throughout the text, including necessary modifications to the title.

We greatly appreciate the reviewer’s important comments and suggestion. We agree that ieCtnnb1’s direct effect on the canonical Wnt signaling is to regulate the transcription of Ctnnb1 in the intestinal epithelia. Therefore, knockout of ieCtnnb1 leads to compromised expression of Ctnnb1 and, consequently, reduced Wnt signaling. The term ‘undermined’ is indeed too strong and has been revised to ‘compromised’ in the revision (line 237). Similar revisions have been made throughout the manuscript. Particularly, the title was changed into ‘A Ctnnb1 enhancer transcriptionally regulates Wnt signaling dosage to balance homeostasis and tumorigenesis of intestinal epithelia’. However, as we state in the following point, decreased levels of β-catenin on ieCtnnb1 loss could lead to indirect effect, including the reduced expression of Bambi, which might cause a more significant decrease of nuclear β-catenin.

(2) It is unclear how the reduction of Ctnnb1 mRNA caused by deletion of ieCtnnb1 in mice could lead to a preferential decrease of nuclear more than membranous β-catenin (Fig. 1K and L). This might reflect a general cell autonomous reduction in Wnt signaling activation; yet, it is not clear how this could occur. Do the authors have any explanations for this?

It's a very important question. We observed that in inCtnnb1 knockout epithelia, the expression of Bambi (BMP and activin membrane-bound inhibitor) was significantly downregulated. Since BAMBI has been reported to stabilize β-catenin and facilitate its nuclear translocation, it is likely that the reduced level of BAMBI resulting from the loss of ieCtnnb1 further decreased nuclear βcatenin. In the revision, the expression change of Bambi has been added in Figure 1M. Moreover, the related content was extensively discussed with proper citations: “We noticed that after knocking out ieCtnnb1, the level of βcatenin in the nuclei of small intestinal crypt cells of Ctnnb1Δi.enh mice decreased more significantly compared to that in the cytoplasm (49.5% vs. 29.8%). Although the loss of ieCtnnb1 should not directly lead to reduced nuclear translocation of β-catenin, RNA-seq results showed that the loss of ieCtnnb1 causes a reduction in the expression of Bambi (BMP and activin membranebound inhibitor), a target gene in the canonical Wnt signaling pathway (Figure 1M). BAMBI promotes the binding of Frizzled to Dishevelled, thereby stabilizing β-catenin and facilitating its nuclear translocation (Lin et al., 2008; Liu et al., 2014; Mai et al., 2014; Zhang et al., 2015). Thus, it is likely that the decreased level of BAMBI resulting from the loss of ieCtnnb1 further reduced nuclear βcatenin”.

(3) In Figure 1 K-L the authors show β-catenin protein level. Why not show its mRNA?

The mRNA levels of Ctnnb1 in small and large intestinal crypts were shown in Figure 1I and 1J, demonstrating reduced expression of Ctnnb1 upon ieCtnnb1 knockout. We hope the reviewer understands that it is unnecessary to measure the nuclear and cytosolic levels of Ctnnb1 transcripts, as the total mRNA level generally reflects the protein level.

(4) Concerning the GSEA of Figure 1 that includes the Wnt pathway components: (a) it would be interesting to see which components and to what extent is their expression affected; (b) why should the expression of Wnt components that are not Wnt target genes be affected in the first place? It is odd to see this described uncritically and used to support the idea of downregulated Wnt signaling.

We appreciate the suggestion and apologize for any lack of clarity. The affected components of the Wnt signaling pathway and the extent of their changes are summarized in Figure 1 – figure supplement 3. Additionally, we have provided explanations for their downregulation. For instance, the reduced expression of Wnt3 and Wnt2b ligands in ieCtnnb1-KO crypts may be attributed to the decreased numbers of Paneth cells.

(5) In lines 251-252 the authors refer to "certain technical issues" in the isolation of cell type from the intestinal epithelium. Why this part should be obscure in the characterization of a tissue for which there are several established protocols of isolation and analysis is not clear. I would rather describe what these issues have been and how they protocol of isolation and analysis is not clear. I would rather describe what these issues have been and how they might have affected the data presented.

We thank the reviewer for pointing this out. The single-cell preparation and sequencing of small intestinal cryptal epithelial cells were carried out largely according to reported protocols with slight modification. The enrichment of live crypt epithelial cells (EpCAM+DAPI-) by flow cytometry and cell filtering after single-cell sequencing were appropriate (Figure 2 – figure supplement 1A1C). We would like to emphasize a few points: (1) Unlike other protocols, we did not exclude immune cells, erythrocytes, or endothelial cells using negative sorting antibodies. (2) When defining cell populations, we focused exclusively on epithelial cell types and did not consider other cell types, such as immune cells. As a result, the so-called “undefined” cells include a mixture of nonepithelial cells. Indeed, markers for erythrocytes (AY036118/Erf1, PMID:12894589) and immune cells (Gm42418 and Lars2, PMID:30940803, PMID: 35659337) were the top three enriched genes in the “undefined” cluster (Figure 2 – figure supplement 1D). (3) Nonetheless, the overall findings remain robust, as key observations such as the loss of Paneth cells and reduced cell proliferation were validated through histological studies. This information has been incorporated into the revised manuscript with related references cited (lines 254-259).

(6) It is interesting that human SNPs exist that seem to fall within the ieCTNNB1 enhancer and affect the gastrointestinal expression of CTNNB1. Could the author report or investigate whether this SNP is present in human populations that have been considered in large-scale studies for colorectal cancer susceptibility? It seems to me a rather obvious next step of extreme importance to be ignored.(7) From Figure 5A a reader could conclude that colorectal tumor cells have a higher expression of CTNNB1 mRNA than in normal epithelium. This is the first time I have seen this observation which somewhat undermines our general understanding of Wnt-induced carcinogenesis exclusively initiated by APC mutations whereby it is β-catenin's protein level, not expression of its mRNA, of crucial importance. I find this to be potentially the most interesting observation of the current study, which could be linked to the activity of the enhancer discovered, and I suggest the authors elaborate more on this and perhaps consider it for future experimental follow-ups.

We appreciate the comments and suggestions. We therefore added related content in the revision (lines 470-475): “Importantly, ieCTNNB1 displayed higher enhancer activity in most CRC samples collected in the study. Moreover, the SNP rs15981379 (C>T) within ieCTNNB1 is associated with the expression of CTNNB1 in the GI tract. Future population studies could investigate how the enhancer activity of ieCTNNB1 and this particular SNP are associated with CRC susceptibility and prognosis”.

(8) I am surprised that the authors, who seem to have dedicated lots of resources to this study, are satisfied by analyzing their ChIP experiments with qPCR rather than sequencing (Figure 6). ChIP-seq would produce a more reliable profile of the HNF4a and CREB1 binding sites on these loci and in other control regions, lending credibility to the whole experiment and binding site identification. Sequencing would also take care of the two following conceptual problems in primer design.First: while the strategy to divide enhancer and promoter in 6 regions to improve the resolution of their finding is commendable, I wonder how the difference in signal reflects primers' efficiency rather than HNF4/CREB1 exact positioning. The possibility of distinguishing between regions 2 and 3, for example, in a ChIP-qPCR experiment, also depends on the average DNA fragment length after sonication, a parameter that is not specified here.Second: what are the primers designed to detect the ieCtnnb1 enhancer amplifying in the yellow-columns samples of Figure 6G? In this sample, the enhancer is deleted, and no amplification should be possible, yet it seems that a value is obtained and set to 1 as a reference value.

This is indeed a crucial point, and we fully agree with the reviewer that “ChIP-seq would produce a more reliable profile of the HNF4a and CREB1 binding sites on these loci and in other control regions”. However, we believe that our current ChIP-qPCR experiments have adequately addressed the potential concerns raised by the reviewers. (1) We have ensured that the DNA fragment length after sonication falls within the range of 200 bp to 500 bp, with an average length of approximately 300 bp (Author response image 1A). We have stated the point in the revised methods section (line 633). (2) We have randomly inspected 14 out of 26 primer sets used in Figure 6 and its supplemental figure (Author response image 1B-E), confirming that all primer sets demonstrate equal amplification efficiency (ranging from 90% to 110%). This information has also been included in the revised methods section (line 650). (3) Figures 6G and 6H show reduced enrichment of HNF4𝛼 (6G) and p-S133-CREB1 (6H) at the Ctnnb1 promoter in ieCtnnb1 knockout ApcMin/+ tumor tissues. The ChIP-qPCR primers used were positioned at the Ctnnb1 promoter, not at ieCtnnb1, with IgG control enrichment serving as the reference values on the Y-axes.

**Author response image 1. sa4fig1:** Quality control assays for ChIP-qPCR. (A) Agarose gel electrophoresis of sonicated DNA. (B-E) Tests of amplification efficiency for primer sets used in ChIP-qPCR.

(9) The ChIP-qPCR showing preferential binding of pS133-CREB1 in small intestinal crypts and CHT15 cells (line 393) should be shown.

The ChIP-qPCR results demonstrating preferential binding of p-S133-

CREB1 over CREB1 have been added in revised Figure 6C, 6D and Figure 6 – Supplement 1C.

(10) It is not entirely clear what the blue tracks represent at the bottom of Figures 6C-D and Figure 6 - Figure Supplement 1C-D. The ChIP-seq profiles of both CREB1 and HNF4a shown in Figures 6A and Figure 6 - Figure Supplement 1A do not seem to match. Taking HNF4a, for example from Figure 6 - Figure Supplement 1A it seems to bind on the Ctnnb1 promoter, while in Figure 6 - Figure Supplement 1D the peaks are within the first intron. I realize this might all be a problem with a different scale across figure panels, but I suggest producing a cleared figure.

We apologize for the confusion. We have revised Figure 6C-6D, Figure 6 - figure supplement 1C-D, and the corresponding legends to enhance clarity. (1) The top panels of Figures 6C and 6D respectively highlight shaded regions of ieCTNNB1 (pink) and the CTNNB1 promoter (grey) in Figure 6A, emphasizing the enrichment of p-S133-CREB1. (2) The top panels of Figure 6 – figure supplement 1C and 1D respectively highlight shaded regions of ieCtnnb1 (pink) and the Ctnnb1 promoter (grey) in Figure 6A – figure supplement 1A, emphasizing the enrichment of HNF4α. (3) Because Figures 6C-6D and Figure 6 - figure supplement 1C-1D respectively correspond to human and mouse genomes, the positions of peaks and scales differ.

(11) In the intro the authors refer to "TCF-4". I suggest they use the more recent unambiguous nomenclature for this family of transcription factors and call it TCF7L2.

TCF-4 has been changed into TCF7L2 in the revision (line 81)

(12) In lines 121-122, the authors write "Although numerous putative enhancers...only a fraction of them were functionally annotated". To what study/studies are the authors referring? Please provide references.

References were added in the revision (line 124)

(13) In some parts the authors use strong words that should in my opinion be attenuated. Examples are: (i) at line 224, "maintains" would be better substituted with "contribute", as in the absence of ieCtnnb1, Ctnnb1 is still abundantly expressed; (ii) at line 266 "compromised" when the proliferative capacity of CFCs and TACs seems to be only mildly reduced; (iii) at line 286 "disrupts", the genes are simply downregulated.

We thank these great suggestions. (1) On lines 224-225, the sentence was revised to: “These data suggest that ieCtnnb1 plays a specific role in regulating the transcription of Ctnnb1 in intestinal epithelia”. (2) On line 271, “compromised” were replaced with “mildly reduced”. (3) In ieCtnnb1 knockout epithelial cells of small intestine, genes related to secretory functions were decreased, while genes related to absorptive functions were increased. Therefore, the term 'disrupts' is more appropriate than 'downregulates'.

**Reviewer #3:**
Line 81, c-Myc should be human MYC (italics) to agree with the other human gene names in this sentence.

c-Myc has been changed into MYC in the revision (line 82)

Line 215, wildtype should be wild-type.

“wildtype” has been changed into “wild-type” in the revision (line 215)

Line 224, Elimination of the enhancer did not abolish expression of Ctnnb1; therefore, it would be better to say that it "helps to maintain Ctnnb1 transcription"

The sentence was changed into “These data suggest that ieCtnnb1 plays a specific role in regulating the transcription of Ctnnb1 in intestinal epithelia” in revision (lines 224-225)

Line 228, perhaps "to activate transcription" is meant.

“active” has been changed into “activate” in the revision (line 228)

Line 235, consider "reduced" instead of "undermined".

“undermined” has been replaced with “compromised” in the revision (line 237)

Line 262, "em" dashes should be a both ends of this insertion.Line 298, "dysfunctional" would be better.Line 356, "samples were".Line 481, 12-hr (add hyphen).

All above points have been optimized according to the reviewer’s suggestion.

Line 712, Is "poly-N" meant?

“Poly-N” indicates undetected bases during sequencing. This explanation was added in the revision (lines 759-760).

Figure 1K, the GAPDH signal is not visible and that panel is unnecessary as there is an H3 control.

Figure 1K and 1L respectively show levels of nuclear and cytoplasmic βcatenin. GAPDH and H3 were used as internal references for the cytoplasmic and nuclear fractions, respectively, confirming both robust fractionation and equal loading.